# TRAINING LARGE LANGUAGE MODELS TO REASON IN PARALLEL WITH GLOBAL FORKING TOKENS

**Sheng Jia**[1,3][†] **Xiao Wang**[2][*] **Shiva Prasad Kasiviswanathan**[2][*]
[1]University of Toronto, [2]Amazon, [3]Vector Institute
{sheng.jia@mail.utoronto.ca, waxao@amazon.com, kasivisw@gmail.com}

 **Code**: https://github.com/Sheng-J/SSFT  🤗 **HF**: shengjia-toronto/ssft

## ABSTRACT

Although LLMs have demonstrated improved performance by scaling parallel test-time compute, doing so relies on generating reasoning paths that are both diverse and accurate. For challenging problems, the forking tokens that trigger diverse yet correct reasoning modes are typically deep in the sampling tree. Consequently, common strategies to encourage diversity, such as temperature scaling, encounter a worsened trade-off between diversity and accuracy. Motivated by this challenge, we treat parallel reasoning as a set-of-next-token-prediction problem and incorporate a set-based global loss into Supervised Fine-Tuning (SFT) using bipartite matching between global forking tokens and unique reasoning traces. We observe that whereas naive fine-tuning with multiple reasoning traces collapses these unique reasoning modes, our proposed method, Set Supervised Fine-Tuning (SSFT), preserves these modes and produces emergent global forking tokens. Global Forking Policy Optimization (GFPO) leverages these maximally steerable tokens to incentivize complex reasoning, and the resulting models consistently outperform their SFT counterparts with GRPO on both math reasoning and execution-based code generation benchmarks.

## 1 INTRODUCTION

Large language models have recently improved reasoning by allocating more test-time compute to generate more tokens before producing the final answer (OpenAI, 2025). However, extended sequential scaling can lead to "overthinking", where performance decreases beyond a certain sequence length (Ghosal et al., 2025; Chen et al., 2024a). To mitigate this, another scaling dimension based on repeated parallel sampling and aggregation (Wang et al., 2022; Brown et al., 2024) has shown success in further boosting reasoning performance. However, these methods rely on LLMs generating diverse yet correct solutions; as tasks become harder, a mechanism for increasing diversity is required. Recent work shows that only a minority of tokens in Chain-of-Thought reasoning (Wei et al., 2022) can act as forking tokens that lead to distinct reasoning modes (Wang et al., 2025b), so as the problem becomes harder and the generation becomes longer, it can become substantially harder to sample them. Also, common practices to encourage diversity, typically through temperature scaling, inherently entail a diversity-accuracy trade-off, as the forking tokens that trigger diverse yet correct reasoning modes are typically located deeply within the sampling tree. Moreover, recent theoretical work also shows that increasing the temperature alone does not necessarily guarantee greater diversity unless the model is explicitly trained to ensure coverage. (Verine et al., 2025).

Building on these observations, we aim to leverage diverse reasoning traces to train for coverage (Guo et al., 2025; Google, 2025b). We introduced *global forking tokens* prior to generate parallel reasoning traces and frame parallel reasoning as a *set prediction* problem. Specifically, given a question, an LLM, conditioned on a reserved set of tokens in a chosen ordering, generates M reasoning sequences in parallel, each aligned with one of M ground-truth reasoning traces. For each ordering, we compute the total autoregressive loss across the generated sequences. By enumerating all possible orderings, we identify the minimum loss, which defines the set language modeling

---

[†]Work done during an internship at Amazon Science.
[*]Equal advising.

loss, conditioned on the distinct forking tokens (Equation 3). This formulation naturally incorporates coverage into the training objective and is capable of learning *global forking tokens* that can serve as prompts to trigger reasoning modes that are both diverse and accurate. We operationalize this idea through our Set Supervised Fine-Tuning (SSFT) framework. Our main contributions are summarized below.

- We introduce global forking tokens and incorporate a set-based loss into SFT via bipartite matching between reserved control tokens and diverse traces (Section 2, Figure 1). The SSFT fine-tuned model globally steer reasoning modes from a single global forking token, reducing dependence on sampling forking tokens mid-generation (Section 2.4; Figure 2).

- We show that SFT on diverse reasoning traces collapses control tokens into a single reasoning mode (Figure 5), whereas the global forking tokens learned by SSFT initiate distinct reasoning. This is visualized quantitatively through discrepant reasoning effort and accuracy over different control tokens (Figure 4) and improved $\text{Pass}@k$ (Figure 3), and qualitatively through distinct math reasoning strategies across forking tokens (Section A.10).

- We demonstrate that, across multiple reasoning benchmarks, Global Forking Policy Optimization can leverage the learned global forking tokens to incentivize diverse complex reasoning without collapsing them, and the resulting models consistently outperform their SFT counterparts with GRPO, improving $\text{Pass}@1$ and $\text{Cons}@k$ (Section 3, Table 1, 2).

## 2 LEARNING GLOBAL FORKING TOKENS VIA SET SUPERVISED FINETUNING

**Background on Language Modeling with Reasoning.** In language modeling, the goal is to train a model $\pi_{\boldsymbol{\theta}}$ to approximate the joint distribution over a sequence of word tokens $\mathbf{x} = \{x_i\}_{i=1}^{\text{T}} \in \mathcal{V}^{\text{T}}$, where T is the sequence length, and each token is within a finite vocabulary set $\mathcal{V}$. An autoregressive model uses the chain rule to represent it as a product of conditionals on the preceding tokens: $\pi_{\boldsymbol{\theta}}(\mathbf{x}) = \prod_{t=1}^{\text{T}} \pi_{\boldsymbol{\theta}}(\mathbf{x}_t|\mathbf{x}_{<t})$. This is known as *next-token-prediction* (NTP) (Radford et al., 2019). For reasoning tasks, we break the sequence of word tokens into: (1) an input prompt $\mathbf{x} = \{x_t\}_{t=1}^{\text{T}_{\mathbf{x}}}$, (2) a special token (or a sequence of tokens) g that initiates reasoning, and (3) a reasoning trace plus the final answer $\mathbf{r} = \{r_t\}_{t=1}^{\text{T}_{\mathbf{r}}}$. To simplify notation, we combine a verifiable answer and a reasoning trace. A reasoning model autoregressively generates a reasoning path and the final answer: $\pi_{\boldsymbol{\theta}}(\mathbf{r}|\mathbf{x}, \mathbf{g}) = \prod_{t=1}^{\text{T}_{\mathbf{r}}} \pi_{\boldsymbol{\theta}}(r_t|\mathbf{x}, \mathbf{g}, \mathbf{r}_{<t})$. To train a reasoning model, Supervised Fine-tuning (SFT) minimizes the negative log-likelihood of a ground-truth reasoning trace, i.e., $\mathcal{L}(\boldsymbol{\theta}) = -\mathbb{E}_{\mathbf{x},\mathbf{r}}[\sum_t \log \pi_{\boldsymbol{\theta}}(r_t|\mathbf{x}, \mathbf{g}, \mathbf{r}_{<t})]$.

### 2.1 PARALLEL REASONING AS SET OF NEXT TOKEN PREDICTION

**Problem Setup.** In this paper, our goal is not only to instill new reasoning capabilities into a model, but also to ensure that prompting with a set of reserved special tokens, in parallel with a question, elicits distinct reasoning traces. We call these *global forking tokens* $\boldsymbol{g} := \{g^{(i)}\}_{i=1}^{\text{N}}$ instantiated as $\{\texttt{<think i>}\}_{i=1}^{\text{N}}$ tags. We use $g^{(i)}$ interchangeably with $\texttt{<think i>}$, depending on context for clarity. We consider a setting with multiple sources of reasoning traces, obtained at low cost without human annotation by distilling from diverse teachers, sampling repeatedly, and potentially filtering with a verifiable metric such as correctness. We adopt this low-cost regime to highlight the effectiveness of our algorithm, though the method extends naturally to settings with well-annotated, human-labeled data. So our problem is to (1) learn to do a set of next-token predictions on multiple distinct yet correct reasoning traces $\mathbf{R} := \{\mathbf{r}^{(j)}\}_{j=1}^{\text{M}}$ in parallel for an input prompt $\mathbf{x}$, and (2) ensure that distinct global forking tokens can uniquely initiate these distinct traces.

To do this, we make a simple change to the NTP loss, which now has two requirements: **(1) Permutation-invariance:** It should not depend on the order of elements in $\mathbf{R}$ and $\boldsymbol{g}$, so we don't penalize a trace that incurs high NTP loss under one forking token if the model predicts it well when conditioned on another. **(2) No shared global forking token:** We want $\{g^{(i)}\}_{i=1}^{\text{N}}$ to uniquely initiate distinct reasoning traces, so this requirement prevents conditioning on the same $g^{(i)}$ when generating distinct traces given a question.

To satisfy these requirements, we incorporate a subproblem in language modeling: finding the minimum cost *bipartite matching configuration* between the left vertices $\{g^{(i)}\}_{i=1}^{\text{N}}$ and the right vertices

$\{\mathbf{r}^{(j)}\}_{j=1}^{M}$ where the cost of each edge between a left vertex $i$ and a right vertex $j$, is the NTP loss of $\mathbf{r}^{(j)}$ conditioned on $g^{(i)}$ and an input prompt $\mathbf{x}$. A *matching configuration* is a set of edges connecting the left and right vertices where no two edges share a common vertex. Without loss of generality, we assume $N \geq M$ to simplify our notation. The total cost involves all vertices on the smaller side of the bipartite graph, so this allows us to write the summation from 1 to $\min\{N, M\}$, which equals $M$ under this assumption. We denote a matching configuration as a finite map $\boldsymbol{\sigma} : \{1, ..., M\} \to \{1, ..., N\}$ such that $\boldsymbol{\sigma}(j) = i \iff \mathbf{r}^{(j)}$ is paired with $g^{(i)}$. Let $\mathfrak{S}_{P} := \{\sigma_k\}_{k=1}^{P}$ denote all the $\binom{N}{M} \times M!$ configurations of a bipartite graph. The total cost of each configuration represents the compatibility between $\{\mathbf{r}^{(j)}\}_{j=1}^{M}$ and $\{g^{(i)}\}_{i=1}^{N}$ under this unique matching. Figure 1 shows an example of a matching configuration.

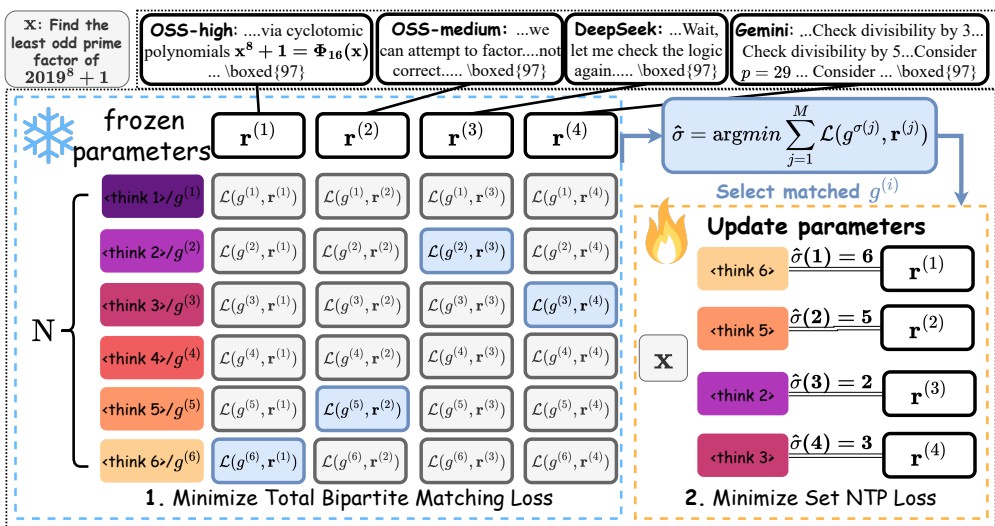

Figure 1: An illustration of one SSFT training step. **Step 1:** We first construct the cost matrix by evaluating all pairwise combinations: for each $\mathbf{r}^{(j)} \in \{\mathbf{r}^{(1)}, \mathbf{r}^{(2)}, \mathbf{r}^{(3)}, \mathbf{r}^{(4)}\}$ and each $g^{(i)} \in \{g^{(1)}, g^{(2)}, g^{(3)}, g^{(4)}, g^{(5)}, g^{(6)}\}$, we compute the NTP loss of $\mathbf{r}^{(j)}$ conditioned on $g^{(i)}$ (Equation (2)). Then we use Hungarian algorithm to find $\hat{\boldsymbol{\sigma}}$ that minimizes the total bipartite matching cost. Here, this minimum is the sum of the losses highlighted in blue, which means $\hat{\boldsymbol{\sigma}} = \{(g^{(6)}, \mathbf{r}^{(1)}), (g^{(5)}, \mathbf{r}^{(2)}), (g^{(2)}, \mathbf{r}^{(3)}), (g^{3}, \mathbf{r}^{(4)})\}$. **Step 2:** We optimize $\boldsymbol{\theta}$ by backpropagating the set of NTP losses for $\mathbf{r}^{(j)}$, each conditioned on $g^{(\hat{\boldsymbol{\sigma}}(j))}$.

## 2.2 SSFT: Minimizing Set of NTP Losses under Optimal Bipartite Matching

Under this formulation, we propose Set Supervised Fine-Tuning (SSFT), which performs two operations at each training step: (1) find the minimum-cost bipartite matching that is optimal for uniquely initiating different reasoning traces (2) and then minimize the NTP losses under the matching configuration to instill diverse reasoning modes conditioned on the matched global forking tokens. We show our implementation in Algorithm 1.

For the first step, we first compute all the entries in the cost matrix such as the one in Figure 1 and then apply the Hungarian algorithm (Kuhn, 1955) to efficiently find the optimal $\hat{\boldsymbol{\sigma}}$:

$$\hat{\boldsymbol{\sigma}} = \arg\min_{\boldsymbol{\sigma} \in \mathfrak{S}_{P}} \sum_{j=1}^{M} \mathcal{L}_{\text{matching}}\left(g^{(\boldsymbol{\sigma}(j))}, \mathbf{r}^{(j)}\right), \quad \text{where} \tag{1}$$

$$\mathcal{L}_{\text{matching}}\left(g^{(i)}, \mathbf{r}^{(j)}\right) = -\text{sg}\left(\frac{1}{T_{\mathbf{r}}} \sum_{t=1}^{T_{\mathbf{r}}} \log \pi_{\boldsymbol{\theta}}\left(\mathbf{r}_{t}^{(j)}|\mathbf{x}, g^{(i)}, \mathbf{r}_{<t}^{(j)}\right)\right) \tag{2}$$

As noted in Equation 2, each matching cost in Equation 1 is the negative log-likelihood of $\mathbf{r}^{(j)}$ conditioned on $g^{(\boldsymbol{\sigma}(j))}$ under the current model parameters. Here, $\text{sg}(\cdot)$ is stop-gradient, as the matching process is done by discrete optimization w.r.t. $\boldsymbol{\sigma}$, so we can save VRAM by not storing

intermediate activations. We explicitly indicate that length normalization is done to remove biases toward trace length, so that the matching is driven by semantic content. After solving $\hat{\boldsymbol{\sigma}}$ for each $(\mathbf{x}, \mathbf{R})$, our second step optimizes model parameters $\boldsymbol{\theta}$ by backpropagating on the matching loss in Equation 3. The expectation is replaced by its sample mean over $(\mathbf{x}, \{\mathbf{r}^{(j)}\}_{j=1}^{M})$ in a mini-batch.

**Time complexity.** In practice, we use only the first $L < T_{\mathbf{r}}$ tokens in Equation 2 to compute the matching cost when early tokens already reveal sufficient differences between traces. This $k = \frac{T_{\mathbf{r}}}{L}$ factor reduces the time complexity of matching to $\mathcal{O}(\frac{NM}{k})$. Backpropagation is performed only on the $M$ matched sequences and can use the full $T_{\mathbf{r}}$ length at the same compute by SFT.

$$\mathcal{L}_{\text{Hungarian}}(\boldsymbol{\theta}) = - \mathop{\mathbb{E}}_{\mathbf{x}, \mathbf{R} \sim \mathcal{D}} \left[ \sum_{j=1}^{M} \sum_{t=1}^{T_{\mathbf{r}}} \log \pi_{\boldsymbol{\theta}} \left( \mathbf{r}_t^{(j)} | \mathbf{x}, \mathbf{g}^{(\hat{\boldsymbol{\sigma}}(j))}, \mathbf{r}_{<t}^{(j)} \right) \right] \tag{3}$$

**Remarks.** The resulting model is not the same as a simple routing of the models independently trained with the nonoverlapping subsets of these traces. Firstly, SSFT allows positive transfer in representation learning within $\{\mathbf{r}^{(j)}\}_{j=1}^{M}$ even though they are matched to different $\{\mathbf{g}^{(i)}\}_{i=1}^{N}$. Secondly, it is not optimal to distill reasoning traces from the same fixed sources for every question if the goal is to maximize both diversity and correctness. Our algorithm supports a variable number of target reasoning traces across training steps, with sources that may also change for each $\mathbf{x}$. Thirdly, even if the two sets of traces, $\{\mathbf{r}_a^{(j)}\}_{j=1}^{M}$ for $\mathbf{x}_a$ and $\{\mathbf{r}_b^{(j)}\}_{j=1}^{M}$ for $\mathbf{x}_b$, are from the same sources, their optimal configurations $\hat{\boldsymbol{\sigma}}_a$ and $\hat{\boldsymbol{\sigma}}_b$ can still vary because a teacher model can reason differently under different questions. Lastly, we reserve more global forking tokens than the maximum number of traces ($N > M$), and empirically observe that all the forking tokens are being matched throughout the process. This is because the extra forking tokens can maximally intra-differentiate similar traces.

## 2.3 GFPO: Optimizing the Selection of Global Forking Token

Although SSFT can leverage unbalanced bipartite graphs to learn correlations between questions and forking tokens given sufficient data, collecting additional traces is often unnecessarily expensive once the model already learns distinct reasoning patterns. Instead, we apply a small number of RL steps using only the policy gradients of global forking tokens—an extremely efficient variant of Wang et al. (2025b;a). This *Global Forking Policy Optimization* (GFPO) properly shapes the output distribution of $\mathbf{g}^{(i)}$ conditioned on $\mathbf{x}$. It can be implemented trivially by adding a few lines of Python slicing to any off-the-shelf code (Sheng et al., 2024), since global forking tokens always begin at the same indices of generations. Full generations are used solely to compute the advantage of $\mathbf{g}^{(i)}$, and they do not participate in backpropagation.

## 2.4 Inference with Learned Global Forking Tokens

**Cons@k.** Our inference protocol with parallel test-time compute is to prompt $i$-th response with `<think (i%N)>` and then do majority voting on their answers.

**Pass@1.** To evaluate a single-path generation, we can either (1) apply GFPO and let the model sample the optimal $\mathbf{g}^{(i)}$ based on the question, or (2) prompt $\mathbf{g}^{(i)}$ that is most likely to correlate with a more complex reasoning mode, as determined by a graph-based heuristic. Inspired by enumerating dissimilar bipartite matchings to reveal node-level variation (Blumenthal et al., 2022), we choose the learned $\mathbf{g}^{(i)}$ with largest coverage. Concretely, each time an optimal matching $\hat{\boldsymbol{\sigma}} \in \{\boldsymbol{\sigma}_k\}_{k=1}^{P}$ is computed, we increment a count $c(\boldsymbol{\sigma}_k)$; empirically, only a finite subset $\mathfrak{S}_p := \{\boldsymbol{\sigma}_k\}_{k=1}^{p} \subseteq \mathfrak{S}_P$ continues to accumulate mass late in training, indicating the stable learned matchings. We then take the union of their edges, and select $\mathbf{g}^{(i^{\star})}$ that matched to the largest number of distinct traces based on Equation 4 for Pass@1. Figure 2 shows an example of the matchings learned by aggregating all edges in $\mathfrak{S}_p$. More details

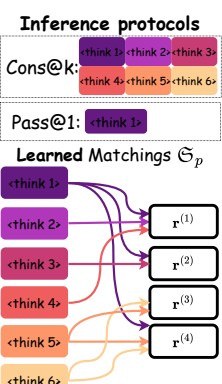

Figure 2: Learned matchings by SSFT-32B in Exp 3, obtained by connecting edges in $\mathfrak{S}_p$.

about $\mathfrak{S}_p{}^1$ are in Appendix A.2.

$$i^\star = \arg\max_i | \bigcup_{\boldsymbol{\sigma} \in \mathfrak{S}_p} \{j | \boldsymbol{\sigma}(j) = i\} | \tag{4}$$

## 3 EXPERIMENTS

We address the following research questions through experiments: **(RQ1)**: In terms of $\mathrm{Pass@1}$ and $\mathrm{Cons@}k$ accuracy, how does a model trained with SSFT perform on reasoning benchmarks? **(RQ2)**: Does finding the optimal bipartite matching matter in reasoning performance? **(RQ3)**: Does training with diverse reasoning traces yield better accuracy and coverage under SSFT compared to standard SFT with temperature scaling? **(RQ4)**: Does prompting with distinct $\{g^{(i)}\}_{i=1}^{N}$ genuinely make a model generate diverse reasoning traces? **(RQ5)**: Is the performance gain from SSFT robust to different fine-tuning datasets and model scales?

### 3.1 EXPERIMENT SETUP

**Training Dataset.** We use the 1,000 questions from s1k dataset Muennighoff et al. (2025). In addition to the R1 (Guo et al., 2025) and Gemini Flash (Google, 2024) traces provided by s1, we also use Claude Opus 4.0/4.1 (Anthropic, 2025) and GPT-OSS-120B (Agarwal et al., 2025) with high and medium reasoning effort to obtain a pool of distilled targets for the 1,000 questions. For each question, we generate two traces per source to populate the pool. We then sample four traces from this pool. We call this s1k-4mixed-reasoning dataset. For additional GFPO, we use 1280 questions from DAPO-Math-17k (Yu et al., 2025).

**Training Details.** We fine-tune Qwen2.5-32B-Instruct (Yang et al., 2025a) for six epochs with a context length of 32,768. We reserve $N = 6$ global forking tokens and use $M = 4$ targets per question. To find the optimal bipartite matching for each input prompt, we consider only the first 1,000 tokens when computing the matching cost in Equation 2. This number is derived from $\lfloor 32768/(MN) \rfloor$ for computational efficiency, so all matching costs fit in one forward pass without extra VRAM. We call this model SSFT-32B. We also include SSFT but choose a random bipartite matching at each step to fine-tune SSFT-32B (random $\boldsymbol{\sigma}$). Models with an additional RL step use the postfix GRPO/GFPO. Exact details on the pool of diverse distillation targets and selection procedure, as well as training hyperparameters, are provided in Appendix A.9 and Table 9.

**Baselines.** Our baselines include two groups: **(Single-Target $\triangle$)** models trained with one trace per question and **(Multi-Target $\bigstar$)** models trained with four traces per question. For **(Single-Target)**, we include s1.1-32B (Muennighoff et al., 2025), which uses 1k DeepSeek-R1 traces. We also fine-tuned an SFT-OSS-distill-32B baseline that trains only on the 1k GPT-OSS traces with high reasoning effort, as these traces achieved the highest correctness on the 1k questions based on an evaluation by Claude 3.5 Sonnet comparing each attempt against the reference answer. For **(Multi-Target)**, we use s1k-4mixed-reasoning to fine-tune SFT-mixed-distill-32B-tags using standard SFT, where each question is duplicated for its distilled traces and `<think i>` is sampled without replacement for each duplicate, treating the four traces as four individual data points. We also include Multiverse-32B (Yang et al., 2025b), which prompts Gemini 2.5-Pro (Google, 2025a) to transform 1k sequential CoTs into parallel CoTs as their training data.

**Evaluation Setup.** Our evaluation tasks consist of AIME24/AIME25 (Ye et al., 2025), MATH-500 (Hendrycks et al., 2021), GPQA-Diamond (Rein et al., 2024), and LiveCodeBench (LCB), where LCB is considered out-of-distribution relative to the fine-tuning dataset. We use LightEval (Habib et al., 2023) as our evaluation framework with generation configurations: temperature=0.7 used in (Guha et al., 2025), top_p=0.95, max length=32768. For $\mathrm{Pass@1}$ accuracy without any parallel test-time compute, we select learned $g^{(1)}$ for SSFT-32B and $g^{(4)}$ for SSFT-32B (random $\boldsymbol{\sigma}$) based on Equation 4. We let the model sample $g^{(i)}$ for SSFT-32B-GFPO. For each $\mathrm{Pass@1}$ accuracy, we compute the average performance over 32 generations. For $\mathrm{Cons@6}$, which applies each of the six global forking tokens once in our method and uses six generations for the baselines, we compute the average over 11 sets of generations to reduce variance in the results. We refer to this as *Pass@1 of Native Cons@6* using a similar terminology as the concurrent work (Wen et al., 2025). Appendix A.10 presents an example of the parallel generations.

---

[1]We choose the subscripts $p$ and P to emphasize that $\mathfrak{S}_p$ is a subset of $\mathfrak{S}_P$.

Table 1: Performance of SSFT compared to baselines on reasoning tasks, reported at Pass@1, Cons@6, and Cons@32. SSFT selects `<think 1>` for Pass@1 and replaces 6 generations with 6 generations prompted by distinct `<think i>` for Cons@$k$. SSFT-GFPO samples `<think i>` for Pass@1. We observe consistent improvements over (i) SFT-OSS-distill-32B, which uses the 1k OSS-high traces; (ii) SFT-mixed-distill-32B-tags, which uses four mixed traces but treats them as individual data; and (iii) SSFT-32B (random $\sigma$), which optimizes Equation 3 with a random $\sigma$.

| | AIME 2024 | AIME 2025 | MATH-500 | GPQA-D | LCB(v5)* |
|---|---|---|---|---|---|
| *Pass@1: Average performance of individual generations (* denotes out-of-distribution tasks.)* | | | | | |
| Qwen2.5-32B-Instruct △ | 15.80 | 10.40 | 80.40 | 47.00 | 23.35 |
| s1.1-32B △ | 54.79 | 44.27 | **92.16** | 62.12 | - |
| Multiverse-32B ★ | 53.80 | 45.80 | 91.80 | 60.70 | - |
| SFT-OSS-distill-32B △ | 57.82 | 48.75 | 89.54 | 60.06 | 34.13 |
| SFT-mixed-distill-32B-tags ★ | 58.23 | 51.96 | 88.49 | 59.96 | 32.34 |
| **SSFT-32B (random $\sigma$) ★** | 61.77 | 55.10 | 89.95 | **62.28** | 35.33 |
| **SSFT-32B ★** | **64.06** | **58.13** | 90.02 | 60.39 | 38.92 |
| **SSFT-32B-GFPO ★** | **64.22** | **58.80** | 89.90 | 62.48 | **42.10** |
| *Pass@1 of Native Cons@6: Average performance of majority voting with 6 parallel generations* | | | | | |
| s1.1-32B △ | 70.30 | 53.33 | 95.60 | 61.45 | - |
| SFT-OSS-distill-32B △ | 72.12 | 65.45 | 95.47 | 61.52 | - |
| SFT-mixed-distill-32B-tags ★ | 73.94 | 70.00 | 95.88 | 58.75 | - |
| **SSFT-32B (random $\sigma$) ★** | 73.03 | 67.58 | 95.67 | 61.87 | - |
| **SSFT-32B ★** | **75.45** | **73.94** | **96.47** | **63.05** | - |
| **SSFT-32B-GFPO ★** | 76.67 | 78.48 | 95.73 | 60.94 | **-** |
| *Cons@32: Majority voting performance with large number of parallel generations* | | | | | |
| s1.1-32B △ | 73.33 | 63.33 | 94.80 | 60.61 | - |
| SFT-OSS-distill-32B △ | 76.66 | 73.33 | 96.00 | 61.60 | - |
| SFT-mixed-distill-32B-tags ★ | 76.67 | 76.67 | 96.20 | 58.59 | |
| **SSFT-32B (random $\sigma$) ★** | 80.00 | 80.00 | 95.60 | **62.63** | - |
| **SSFT-32B ★** | **83.33** | **86.67** | **96.80** | 61.62 | - |
| **SSFT-32B-GFPO ★** | 83.33 | 83.33 | 96.80 | 62.12 | - |

△ indicates training with single-target data and ★ indicates training with multi-target data.

## 3.2 Evaluating SSFT on Reasoning Benchmarks

For **RQ1**, we see in Table 1 that SSFT delivers the best Pass@1 accuracy, 64.06 on AIME24 and 58.13 on AIME25, outperforming SFT-mixed-32B by 8.33% and 6.57%, respectively. We also observe consistent improvements on all four tasks under parallel test-time compute at two scales, Cons@6 and Cons@32, over SFT-mixed-32B, which was trained on the same reasoning traces. Some notable results are Cons@6 = 73.94%, Cons@32 = 86.67% on AIME25. To answer **RQ2**, we observe consistent improvements over SSFT-32B (random $\sigma$), with especially strong gains at Cons@6 on AIME25, where effectiveness with few parallel generations is critical. As shown later in Figures 4 and 5, optimal bipartite matching is essential to prevent collapsing reasoning modes. Results with SSFT-GFPO confirm that GFPO successfully sharpens the output distribution of $g^{(i)}$ conditioned on $x$. We even observe positive gains in Pass@1. Appendix A.3 reports the results for baselines with RL. For out-of-distribution generalization to LCB, SSFT also yields the largest gains. **Coverage.** For **RQ3**, we compare our method against SFT-mixed-32B under various $k$ in Pass@$k$ accuracy with 32 generations to assess generation coverage. Figure 3 shows that SSFT achieves higher coverage across nearly all values of $k$. SFT-mixed-32B requires more allowed attempts and higher temperature to match the coverage of SSFT at the cost of lowering its Pass@1 and Cons@6.

## 3.3 Evaluating Parallel Reasoning Diversity and Learned Matchings

Addressing **RQ4**, we show that our global forking tokens genuinely initiate distinct reasoning traces and offer **a new mechanism** for leveraging test-time compute.

**Emerging Diverse Reasoning Modes.** Using the Cons@6 results in Table 1, we form six sets of generations, each prompted by a distinct $g^{(i)}$. For each set, we show the average accuracy and the

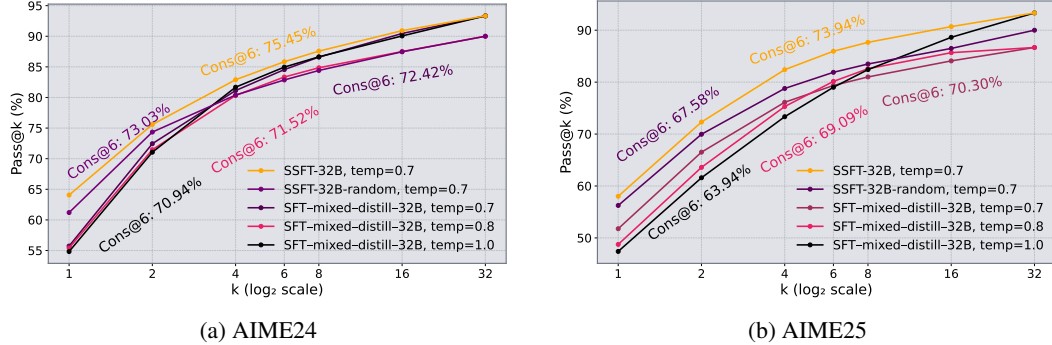

(a) AIME24      (b) AIME25

Figure 3: Coverage of SSFT compared to SFT-mixed-distill-32B with temperature scaling, reported at $\text{Pass}@k$. For convenience, we also report the $\text{Cons}@6$ accuracy next to each line. In AIME25, SFT-mixed-distill-32B needs to raise the inference temperature to 1 and use more attempts to match the coverage at the cost of lowering its $\text{Pass}@1$ and $\text{Cons}@6$ accuracy, further widening the gaps.

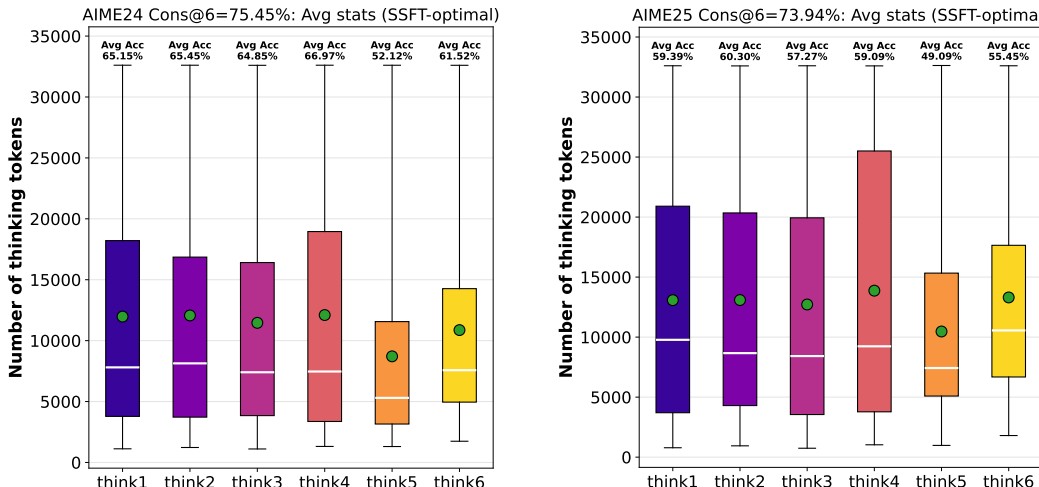

Figure 4: (SSFT, optimal matching). Distribution of thinking-token counts and average performance on AIME24 (left) and AIME25 (right) prompted by a distinct `<think 1>,...,<think 6>`.

distribution of thinking-token counts: Figure 4 for SSFT with optimal bipartite matching and Figure 5 for SSFT with random matching. **Reasoning length:** Length partially indicates the diversity in reasoning, and we see clear differences for SSFT with optimal matching, despite the absence of hand-crafted matching rule or information about these traces. The consistency of these distributions across AIME24 and AIME25 indicates the differences is not from randomness, whereas randomly assigning a `<think i>`, as in concurrent work (Wen et al., 2025), does not yield clear or consistent differences in reasoning length, as shown in Figure 5. **Performance:** After finetuning with random matching, prompting with a distinct `<think i>` shows no meaningful impact ($\approx 61\%$ on AIME24 and $\approx 55\%$ on AIME25). With optimal matching, SSFT elicits distinct reasoning modes initiated by `<think 1>,...,<think 4>` that reach around $65\%$ on AIME24 and $\geq 59\%$ on AIME25, with different lengths. Although `<think 5>` and `<think 6>` are weaker due to shorter reasoning modes, the average of these and, especially, $\text{Cons}@6$ performance improve consistently with them.

**Visualization of Learned Matchings.** We observe in Figure 2 that some $\text{g}^{(i)} \in \boldsymbol{g}$ from SSFT-32B have unique configuration of matched edges with $\{\mathbf{r}^{(j)}\}_{j=1}^{4}$. This is a positive indication that $\{\text{g}^{(i)}\}_{i=1}^{6}$ are likely to initiate distinct reasoning modes. We hypothesize that $\{\text{g}^{(i)}\}_{i=1}^{N}$ can still yield a unique edge-matching configuration even if a subset of $\{\mathbf{r}^{(j)}\}_{j=1}^{4}$ are difficult to distinguish. For interpretability, we fine-tune using the same four teacher models for each $\mathbf{x}$: GPT-OSS high, medium, R1, and Gemini. We call this dataset s1k-4teachers-reasoning dataset, and ask whether SSFT associate a unique $\text{g}^{(i)}$ to the Gemini and R1 traces which have easily identifiable reasoning patterns, and then have the rest matched to OSS-high/OSS-medium traces in differ-

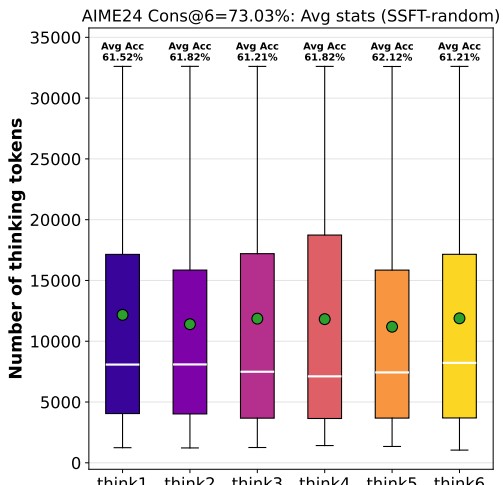
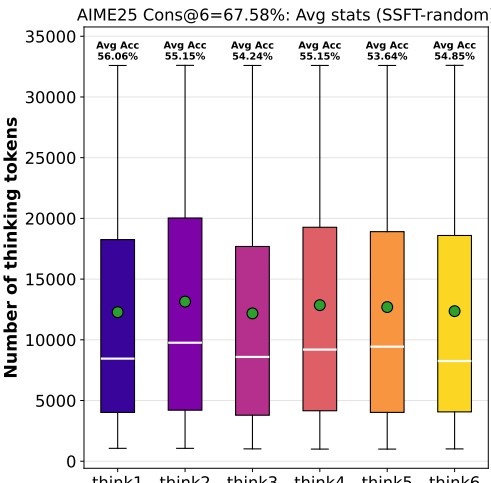

Figure 5: (SSFT, random matching). Distribution of thinking-token counts and average performance on AIME24 (left) and AIME25 (right) prompted by a distinct `<think 1>,...,<think 6>`.

ent ways (i.e. only matched to OSS-high, only matched to OSS-med, and matched to both). We track source indices only for evaluation; the model still receives an unordered set of traces with no source information for each question. We study 3 bipartite matching settings for SSFT. Figure 6 shows the evolution of matchings learned under these 3 hyperparameters SSFT (a) random matching with four reserved $g^{(i)}$, (b) optimal matching with four reserved $g^{(i)}$, and (c) optimal matching with six reserved $g^{(i)}$. Initially, all the configurations $\sigma \in \mathfrak{S}_P$ accumulate mass as there is no correlations between $g$ sand $\{r^{(j)}\}_{j=1}^{M}$. Figure 6a shows SSFT under random matching does not shrink the size of configs computed as optimal, meaning that no correlations are learned between $\{g^{(i)}\}_{i=1}^{4}$ and $\{r^{(j)}\}_{j=1}^{4}$. By contrast, Figure 6b and Figure 6c show the emergence of only a strict subset of matching configurations in $\mathfrak{S}_P$. This indicates some correlations between $\{g^{(i)}\}_{i=1}^{N}$ and $\{r^{(j)}\}_{j=1}^{M}$ are indeed learned through SSFT. We first visualize the learned matchings of the model with four $g^{(i)}$ in Figure 8b. We observe that $g^{(1)}$ and $g^{(2)}$ are uniquely matched to the R1 and Gemini traces, showing that SSFT can indeed uniquely associate $g^{(i)}$ to sufficiently diverse reasoning traces. Now to confirm our previous hypothesis, we see the unique learned matchings $(g^{(3)}, (\text{OSS-high, OSS-med})), (g^{(4)}, \text{OSS-med})$. Furthermore, by connecting all the edges in $\mathfrak{S}_p$ from SSFT with 6 forking tokens (Figure 6c), we also see unique learned matchings $(g^{(3)}, \text{OSS-high}), (g^{(4)}, \text{OSS-med}), (g^{(5)}, (\text{OSS-med, OSS-high}))$ in Figure 8c. This confirms that the global forking tokens can identify unique correlations even among highly similar traces.

# 4 ROBUSTNESS AND ABLATION STUDIES

**Robustness to Fine-tuning with Code Generation traces.** We study whether SSFT can improve Pass@1 on more open-ended tasks by instilling specialized, complex reasoning modes without mode collapse, unlike standard SFT where such modes can be corrupted by merging together. We select 1K coding questions from Open-Thoughts (Guha et al., 2025), and use R1 to generate four reasoning traces per question to construct `code1k-4mixed-reasoning`. We apply SSFT on this dataset, and then GFPO on 1k questions from `Intellect-2-RL` (Team

Table 2: Comparison of SSFT and baseline fine-tuned models trained on code-generation data, evaluated on one coding task and three math reasoning tasks that are out-of-distribution relative to the fine-tuning set. At test time, `<think i>` is sampled by the models.

| | LCB(v5) | AIME24* | AIME25* | MATH500* |
|---|---|---|---|---|
| *Pass@1 w/ 16 samples for LCB. (* denotes OOD w.r.t post-training)* | | | | |
| Qwen2.5-32B-Instruct | 23.35 | 15.80 | 10.40 | 80.40 |
| SFT-mixed-distill-32B-code | 47.13 | 34.69 | 24.17 | 89.39 |
| **SSFT-32B-code (random)** | 45.36 | 39.06 | 32.92 | 89.46 |
| **SSFT-32B-code** | 52.07 | 43.23 | 32.82 | 89.96 |

et al., 2025). From the Pass@1 comparison in Table 2, SSFT better elicits complex reasoning modes of higher accuracy than baselines that do not use a set loss. Full results are in Appendix 5.

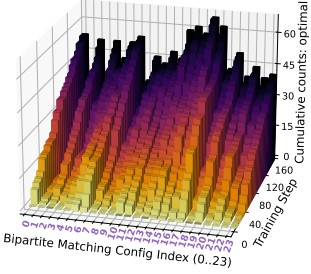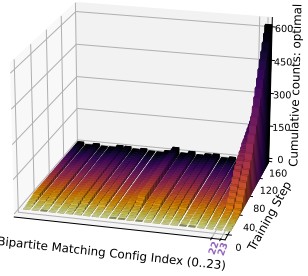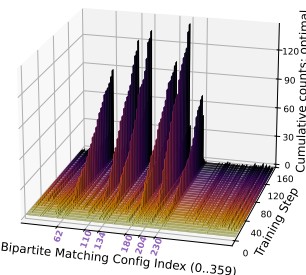

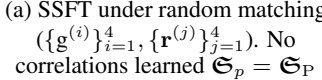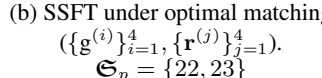

(a) SSFT under random matching ($\{\mathbf{g}^{(i)}\}_{i=1}^4, \{\mathbf{r}^{(j)}\}_{j=1}^4$). No correlations learned $\mathfrak{S}_p = \mathfrak{S}_{\mathrm{P}}$

(b) SSFT under optimal matching ($\{\mathbf{g}^{(i)}\}_{i=1}^4, \{\mathbf{r}^{(j)}\}_{j=1}^4$). $\mathfrak{S}_p = \{22, 23\}$

(c) SSFT under optimal matching ($\{\mathbf{g}^{(i)}\}_{i=1}^6, \{\mathbf{r}^{(j)}\}_{j=1}^4$). $\mathfrak{S}_p = \{62, 110, 134, 180, 204, 230\}$

Figure 6: Cumulative counts of $\boldsymbol{\sigma}_k \in \mathfrak{S}_{\mathrm{P}}$ computed as optimal over training. Note that $\mathfrak{S}_{\mathrm{P}}$ and $\mathfrak{S}_p$ are defined in Sections 2.1 and 2.4, respectively. Front axis: matching configuration index $k$. Depth: training step $t$. Bar height is the cumulative counts. These are the evolution of matchings during training 3 Qwen-32B-Instruct models under 3 bipartite matching settings. In this case study, the $\{\mathbf{r}^{(j)}\}_{j=1}^4$ are always (GPT-OSS-high, GPT-OSS-med, R1, Gemini) for each question. Note that *Random matching* method does not minimize Eqn 3 under optimal matching, but we track it. We observe $\mathfrak{S}_p = \mathfrak{S}_{\mathrm{P}}$ with random matching, meaning no correlations learned. But by optimizing Hungarian loss, we see the emergence of $\mathfrak{S}_p \subset \mathfrak{S}_{\mathrm{P}}$.

**Robustness to Fine-Tuning Data Quality.** To test whether our gains depend on the traces generated by our procedure and on a small high-quality dataset (Muennighoff et al., 2025), we fine-tune on a public dataset that already provides 2-4 reasoning traces per question: the 93k math set of Face (2025). Because this dataset has been successful for fine-tuning Qwen2.5-Math-7B, we adopt that base model and compare SSFT against SFT trained on all available traces. Details are in Appendix A.9.4. Addressing **RQ5**, Table 3 shows consistent improvements in both Pass@1 and Cons@32. The results indicate SSFT is effective for larger public dataset with less diverse traces.

Table 3: Performance of SSFT versus SFT trained solely on publicly available distillation targets. The setup uses the 93k math questions from Face (2025) with Qwen2.5-Math-7B as the base model. SFT-OpenR1-93k-7B uses the same distillation targets as SFT-mixed-distill-32B in Table 1.

| Model | AIME 2024 | | AIME 2025 | | MATH-500 | | GPQA-D | |
|---|---|---|---|---|---|---|---|---|
| | Pass@1 | Cons@32 | Pass@1 | Cons@32 | Pass@1 | Cons@32 | Pass@1 | Cons@32 |
| Qwen2.5-Math-7B-Instruct | 10.42 | 20.00 | 9.48 | 23.33 | 81.87 | 87.40 | 30.29 | 30.30 |
| SFT-OpenR1-93k-7B | 46.15 | 66.67 | 34.17 | 50.00 | 86.62 | 90.20 | 46.35 | 47.98 |
| **SSFT-OpenR1-93k-7B** | **51.25** | **73.33** | **35.52** | **56.66** | **89.74** | **93.60** | **46.86** | **48.90** |

**Robustness to Model Family & Size.** In Appendix A.5, SSFT also shows gains with Qwen3-4B-Base and Llama3.1-8B-Instruct, though the improvements are larger for the bigger Qwen models.

**Impact of Matching Length.** We choose $\mathrm{L} = \frac{\mathrm{max\_seq\_len}}{\mathrm{MN}} \approx 1000$ because it is roughly the largest value that allows NM matching costs to be computed in a single forward pass without extra VRAM. Figure 7 shows this adds very minimal computation because backpropagation and storing activations still dominate the training time. Note that in our setting, $\mathrm{L} = 1000$ was sufficient for maximally differentiating complex reasoning modes and preserving high-accuracy traces, as reflected by Pass@1 and diversity in Cons@$k$ in Figure 7. Increasing L does not hurt Pass@1 or Cons@32. When additional compute time permits, L can be increased further and stopped once performance plateaus.

## 5 RELATED WORK

**Test-time Scaling.** There has been a surge of work fine-tuning LLMs to reason longer, using reinforcement learning for frontier models (OpenAI, 2024; Shao et al., 2024; xAI, 2025; Yang et al.,

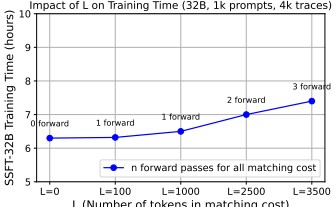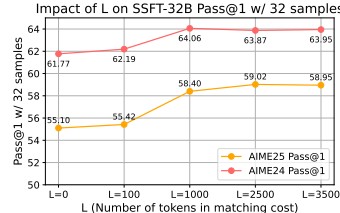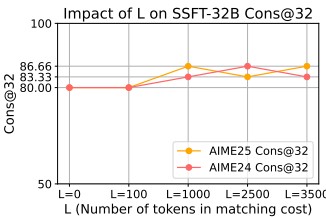

Figure 7: **Left:** Impact on training time. **Middle:** Impact on $\mathrm{Pass@1}$. **Right:** Impact on $\mathrm{Cons@32}$. **Ablation on the number of tokens** L. This is a hyperparameter when computing the matching cost in Eq. 2. Setting $L = 0$ reduces to SSFT with random matching. For each L, we apply SSFT with matching $(N = 6, M = 4)$ to fine-tune Qwen2.5-32B-Instruct on `s1k-4mixed-reasoning` and evaluate training time, Pass@1 on two challenging reasoning tasks, and $\mathrm{Cons@32}$ on the same tasks.

2025a) and supervised fine-tuning for smaller ones (Muennighoff et al., 2025; Hu et al., 2025). These methods enable LLMs to improve reasoning by allocating more test-time compute to sequential, iterative refinement such as self-reflection (Guo et al., 2025; Liu et al., 2025b). However, extended sequential reasoning can be more sensitive to the order of reasoning steps and may result in failures (Chen et al., 2024b), and performance can start to degrade beyond a certain length due to "overthinking" (Ghosal et al., 2025). Our goal is to study the effective use of diverse reasoning traces to fine-tune small language models, essential for agentic AI (Belcak et al., 2025).

**Parallel Reasoning.** Parallel scaling methods such as self-consistency (Wang et al., 2022) and Best-of-N (Lightman et al., 2023) improve LLM performance by generating multiple reasoning paths in parallel and aggregating them. These methods fundamentally require choosing a temperature that can generate diverse reasoning paths, but a recent theoretical work shows that increasing temperature can sometimes fail to increase diversity if language models are not trained towards coverage (Verine et al., 2025). Other search-based methods such as Monte Carlo tree search (MCTS) (Zhang et al., 2024) and Tree of Thoughts (ToT) (Yao et al., 2023) apply heuristic-guided search with an external verifier to do more deliberate search to increase the coverage (Yao et al., 2023; Zhang et al., 2024). However, their dependence on heuristics and domain-specific knowledge can limit their applicable tasks. Regarding training LLMs with parallel reasoning traces, Yang et al. (2025b) proposes training with parallel CoTs decomposed from sequential CoTs, and our concurrent work Wen et al. (2025) proposes to train with multiple reasoning traces distilled from teacher models. These works show native parallel scaling can surpass sequential scaling within certain token limits. However, we aim to show that training on diverse distilled traces with our set language modeling loss enables the model to learn global forking tokens that trigger distinct reasoning modes, improving $\mathrm{Pass@1}$ and $\mathrm{Cons@}k$ over baselines fine-tuned on the same dataset, whether on subsets or the full set.

**Set-based Global Loss in Deep Learning.** DETR introduces end-to-end object detection with a set global loss (Carion et al., 2020; Minderer et al., 2022), whose success in parallel bounding-box prediction inspires our approach. We are the first to extend this to language modeling: while DETR predicts a set of tokens in parallel to match a list of bounding boxes, we predict a set of sequential reasoning paths initiated by global forking tokens and assess the matchings based on autoregressive losses. We adapt the set-based loss with autoregressive (AR) models, rather than diffusion-based models (Nie et al., 2025), because AR models achieve superior reasoning performance.

## 6 CONCLUSION

We show that diverse reasoning traces can be leveraged to learn global forking tokens that globally steer diverse and accurate reasoning modes. We propose Set Supervised Fine-Tuning (SSFT), which employs bipartite matching between such control tokens and traces to compute a set language modeling loss. We visualize that SSFT preserves distinct reasoning modes that SFT collapses, and produces emergent global forking tokens that Global Forking Policy Optimization leverages to incentivize complex reasoning. The resulting models consistently outperform their SFT counterparts with GRPO, improving $\mathrm{Pass@1}$ and $\mathrm{Cons@}k$ across math reasoning benchmarks and $\mathrm{Pass@1}$ on execution-based code generation benchmarks. For future work, we plan to scale up the underlying bipartite graph and explore more general settings beyond multi-teacher distillation.

## ACKNOWLEDGEMENTS

The authors thank Michael Zhang, Silviu Pitis, Andrew Li, and Elena Zhang at the University of Toronto for their valuable feedback. We also acknowledge Amazon Web Services for providing the computing resources that made this research possible.

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

## A   APPENDIX

### A.1   LLM USAGE IN PAPER

We used ChatGPT (OpenAI, GPT-5) in September 2025 for occasional language polishing only. This is done when we really wanted to make sure there are no grammatical errors in a few sentences. No text, code, experiment results, or figures were generated by the LLM. We made our own hypothesis, completed all technical content on our own, and made our conclusions. We also verified all outputs by ourselves. The occasional grammar check is done by typing into LLM chatbox.

### A.2   LEARNED MATCHINGS BETWEEN GLOBAL FORKING TOKENS AND TRACES

Initially, any of the bipartite matching configuration $\boldsymbol{\sigma}_k \in \mathfrak{S}_\mathrm{P}$ can be computed as optimal, as the reserved global forking tokens $\{g^{(i)}\}_{i=1}^\mathrm{N}$ have no specific correlations with these traces $\{\mathbf{r}^{(j)}\}_{j=1}^\mathrm{M}$. This can be observed in Figures 6 and 8 that $c(\boldsymbol{\sigma}_k)$, the count of configuration $\boldsymbol{\sigma}_k$ being optimal during training, uniformly increases on all configuration indexes. However, as training goes with SSFT, we notice only a subset of $\mathfrak{S}_\mathrm{P}$ accumulates mass. This indicates there are some unique correlations learned between $\{g^{(i)}\}_{i=1}^\mathrm{N}$ and $\{\mathbf{r}^{(j)}\}_{j=1}^\mathrm{M}$. This subset is denoted as $\mathfrak{S}_p = \{\boldsymbol{\sigma}_k\}_{k=1}^p$, and we call the unique edges in $\mathfrak{S}_p$ as *learned matchings*.

#### A.2.1   HOW TO CHOOSE THE GLOBAL FORKING TOKEN FOR PASS@1

To find $\mathfrak{S}_p$, we can simply track which configurations $\boldsymbol{\sigma}$ still accumulate mass in the last epoch. Then we can connect all the unique edges in $\mathfrak{S}_p$ to visualize learned matchings. However, multiple global forking tokens may share the maximum number of connected edges in the learned matchings. To break the tie, we treat the counts as edge weights and select $g^{(i)}$ with the largest weighted degree. We provide this implementation in our code.

#### A.2.2   MORE VISUALIZATIONS ON LEARNED MATCHINGS

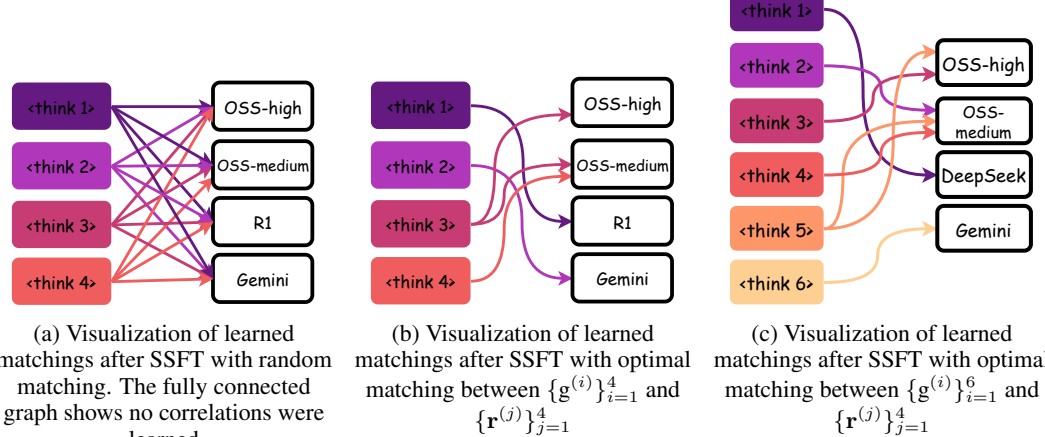

(a) Visualization of learned matchings after SSFT with random matching. The fully connected graph shows no correlations were learned

(b) Visualization of learned matchings after SSFT with optimal matching between $\{g^{(i)}\}_{i=1}^4$ and $\{\mathbf{r}^{(j)}\}_{j=1}^4$

(c) Visualization of learned matchings after SSFT with optimal matching between $\{g^{(i)}\}_{i=1}^6$ and $\{\mathbf{r}^{(j)}\}_{j=1}^4$

Figure 8: These learned matching visualizations are obtained by connecting edges in the subset of configurations $\{\boldsymbol{\sigma}_k\}_{k=1}^p$ that still accumulate mass towards the end of training in Figure 6. These models are fine-tuned using the same GPT-OSS-high, GPT-OSS-medium, R1, and Gemini traces for each question, so we can better interpret the learned matchings

### A.3   GLOBAL FORKING POLICY OPTIMIZATION FOR Pass@1 INFERENCE BY SAMPLING $g^{(i)}$.

For single-path generation, we prompt the $g^{(i)}$ most likely to correlate with a complex reasoning mode using a graph-based heuristic that selects the $g^{(i)}$ matched to the largest number of distinct reasoning traces. Since theoretically justifying any single heuristic is difficult, we apply a small amount of RL to learn the optimal distribution over global forking tokens conditioned on each

question. Using cold-start SFT before RL is common practice in LLM post-training (Guo et al., 2025), and recent work shows that RLVR alone does not incentivize new capabilities (Yue et al., 2025; Liu et al., 2025a).

**Setup.** We use 1,280 questions from `DAPO-Math-17k` and run GRPO for 10 steps using only the policy gradients of $g^{(i)}$. These are referred to as GFPO. We also let a baseline optimize full tokens even though that is much more computationally expensive. The base SFT models include SSFT-32B, SSFT-32B (random $\sigma$), SFT-mixed-distill-32B-tags, trained using `s1k-4mixed-reasoning`. Hyperparameters are in Table 9.

**Results.** In Table 4, we show that SSFT instills diverse, high-accuracy reasoning modes without mode collapse, and that merely 10 steps of GFPO can shape the output distribution of $g^{(i)}$ conditioned on $\mathbf{x}$. By comparing against baselines with RL, we see that the gains come from the synergy between GFPO and the global forking tokens learned by SSFT with a principled set loss, rather than from spurious rewards (Shao et al., 2025). In our setting, SFT-GRPO that optimizes all tokens still cannot match SSFT-GFPO, despite using substantially more compute.

Table 4: Comparison of SSFT and baselines on three math reasoning tasks after further RL fine-tuning only on global forking tokens, denoted as GFPO. We also include an expensive GRPO baseline that optimizes the full sequences. SFT/SSFT uses s4k-mixed-reasoning data, and GRPO/GFPO uses only 1280 questions from DAPO-Math-17K (Yu et al., 2025). The optimal `<think i>` for each question is sampled by the models in this evaluation. By avoiding reasoning mode collapse observed in standard SFT-mixed-distill and SSFT (random $\sigma$) with random matching, SSFT preserves complex, high-accuracy reasoning modes. Standard GRPO that applies gradients to full sequences does not improve performance and can even lead to a slight drop in this small-scale RL setting.

| | AIME 2024 | AIME 2025 | MATH-500 | LCB(v5)* |
|---|---|---|---|---|
| *Pass@1: Average performance of 32 individual generations (AIME24/25, MATH-500)* | | | | |
| SFT-mixed-distill-32B-tags-GRPO | 58.85 | 52.40 | 88.85 | 37.13 |
| SFT-mixed-distill-32B-tags-GFPO | 59.80 | 54.06 | 88.98 | 37.72 |
| SSFT-32B (random $\sigma$)- GFPO | 59.58 | 53.96 | 89.87 | 36.53 |
| SSFT-32B- GFPO | **64.22** | **58.80** | **89.90** | **42.10** |

## A.4 ROBUSTNESS TO FINE-TUNING WITH CODE GENERATION TRACES

We investigate the advantage of SSFT over SFT when the fine-tuning dataset consists of traces from open-ended domains, such as code generation with unit test cases. We study whether SSFT can improve Pass@1 on competitive coding by instilling specialized coding reasoning modes without mode collapse.

**Setup.** We select 1K coding questions from Open-Thoughts (Guha et al., 2025), use DeepSeek-R1 to generate four reasoning traces per question to construct `code1k-4mixed-reasoning`. We apply SSFT on this dataset, and then GFPO on 1k questions from `Intellect-2-RL` (Team et al., 2025). At test time, each model samples `<think i>` for each question. For evaluation, we use LiveCodeBench(v5) and assess whether the models can generate valid code that passes unit tests when executed. Although the evaluation still relies on a verifiable metric that avoids evaluation bias, the resulting code is far more open-ended than in math tasks, where only a final numeric answer is extracted and scored. We still included the math tasks for OOD evaluation.

**Results.** Table 5 reports the full comparison between SSFT and baseline methods under fine-tuning with code-generation traces. SSFT outperforms the baselines on LiveCodeBench and shows stronger generalization to the math domain, despite being fine-tuned only on code-generation traces. These results show the robustness of SSFT to using traces collected from more open-ended tasks. Although correctness and diversity are less clearly defined for code generation, SSFT instills diverse and specialized reasoning modes, and RL can quickly leverage these to find optimal forking path for each coding task. In fact, the Pass@1 gain on LCB after SSFT with code traces is slightly larger than the gain on math tasks after SSFT with math traces (Table 1). One explanation is that

open-ended tasks involve a much larger search space, and global forking tokens help the model avoid getting stuck in suboptimal reasoning modes.

Table 5: Comparison of SSFT and baselines on one coding benchmark, and three math reasoning tasks that are out-of-distribution relative to the fine-tuning set, since the fine-tuning dataset `code1k-4mixed-reasoning` contains only coding questions and traces. All methods use Qwen2.5-32B-Instruct as the base model. SFT-mixed-distill denotes standard SFT where different traces for the same prompt are treated as independent training examples. SSFT (random $\sigma$) denotes SSFT with a random bipartite matching at each training step. All the three models have gone additional small-scale GFPO for optimal forking selection with 1000 coding questions from `Intellect-2-RL`. At test time, `<think i>` is sampled by the models.

| | LCB(v5) | AIME 2024* | AIME 2025* | MATH-500* |
|---|---|---|---|---|
| *Pass@1: avg over n=16 for LCB, n=32 for math tasks. (* denotes out-of-distribution relative to fine-tuning)* | | | | |
| Qwen2.5-32B-Instruct △ | 23.35 | 15.80 | 10.40 | 80.40 |
| SFT-mixed-distill-32B-code ★ | 47.13 | 34.69 | 24.17 | 89.39 |
| **SSFT-32B-code (random $\sigma$) ★** | 45.36 | 39.06 | 31.56 | 89.46 |
| **SSFT-32B-code ★** | 52.07 | 43.23 | 32.82 | 89.96 |
| *Pass@1 of Cons@6: Average performance of majority voting with 6 parallel generations* | | | | |
| SFT-mixed-distill-32B-code ★ | - | 41.52 | 30.30 | 91.70 |
| **SSFT-32B-code (random $\sigma$) ★** | - | 46.97 | 38.18 | 92.43 |
| **SSFT-32B-code ★** | - | 51.82 | 37.88 | 93.25 |
| *Cons@32: Majority voting performance with large number of parallel generations* | | | | |
| SFT-mixed-distill-32B-code ★ | - | 46.67 | 33.33 | 91.40 |
| **SSFT-32B-code (random $\sigma$) ★** | - | 46.67 | 43.33 | 92.00 |
| **SSFT-32B-code ★** | - | 56.67 | 46.67 | 93.00 |

△ indicates training with single-target data and ★ indicates training with multi-target data.

## A.5 ROBUSTNESS TO MODEL FAMILY AND SIZE

**Setup.** We use Qwen3-4B-Base and Llama3.1-8B-Instruct as additional base models, repeating the main experiment from Section 3. Results are reported in Tables 6 and 7, respectively.

**Results.** Table 6 shows that SSFT remains robust on a smaller 4B model from the newer Qwen3 family (Yang et al., 2025a), and Table 7 shows that it is also robust on an 8B model from the older Llama3.1-8B-Instruct family, achieving higher scores on most reported benchmarks. However, the performance gains are smaller than those observed with the larger Qwen2.5-32B.

## A.6 GENERALIZATION TO DOMAINS UNSEEN DURING SSFT.

**SSFT using math traces → Coding**. We follow Wang et al. (2025b) and study out-of-distribution generalization to code generation using LiveCodeBench(V5) as the evaluation task after fine-tuning with only math data `s1k-4mixed-reasoning`, 1.28k questions from `DAPO-Math-17k`. In Tables 1 and 4, we see that SSFT improves OOD generalization by about 5% over the baselines and about 19% over the base model in the 32B setting. For the 4B model, Table 6 shows that it is about 1% lower than one baseline, but still improves over the base model by about 7%.

**SSFT using coding traces → Math**. Since we fine-tuned models with only code generation traces, we also study the other direction. AIME24/25 and MATH-500 are used as OOD tasks. Table 5 shows that SSFT yields significant improvements over the baselines across all math tasks, indicating that the advantage on LCB transfers to math reasoning. 5

Table 6: Comparison of SSFT with baselines on three math reasoning tasks, where all methods use Qwen3-4B-Base as the base model and are trained on the `s1k-4mixed-reasoning`. SFT-mixed-distill denotes standard SFT where different traces for the same prompt are treated as independent training examples. SSFT (random $\sigma$) denotes SSFT with a random bipartite matching. For Pass@1, SSFT uses a graph-based heuristic: it prompts the forking token, `<think i>`, with the largest number of connected edges in the learned matchings.

| | AIME 2024 | AIME 2025 | MATH-500 | LCB(v5)[*] |
|---|---|---|---|---|
| *Pass@1: Average performance of individual generations. LCB is OOD relative to the fine-tuning data.* | | | | |
| Qwen3-4B-Base★ | 9.79 | 6.56 | 66.34 | 7.19 |
| SFT-mixed-distill-Qwen3-4B-tags★ | 21.46 | 21.98 | 78.76 | **15.57** |
| SSFT-Qwen3-4B (random $\sigma$)★ | 23.33 | 21.87 | 79.63 | 12.57 |
| SSFT-Qwen3-4B★ | **23.44** | **23.13** | **81.85** | 14.37 |
| *Cons@6: Average performance of majority voting with 6 parallel generations* | | | | |
| Qwen3-4B-Base★ | 13.03 | 10.91 | 78.63 | - |
| SFT-mixed-distill-Qwen3-4B-tags★ | 30.61 | 28.18 | 89.67 | - |
| SSFT-Qwen3-4B (random $\sigma$)★ | 30.91 | **28.79** | 89.73 | - |
| SSFT-Qwen3-4B★ | **32.42** | **28.79** | **91.20** | - |
| *Cons@32: Majority voting performance with large number of parallel generations* | | | | |
| Qwen3-4B-Base★ | 20.00 | 16.67 | 83.00 | - |
| SFT-mixed-distill-Qwen3-4B-tags★ | 36.67 | 40.00 | **91.80** | - |
| SSFT-Qwen3-4B (random $\sigma$)★ | **43.33** | 36.67 | 91.60 | - |
| SSFT-Qwen3-4B★ | 40.00 | **43.33** | **91.80** | - |

△ indicates training with single-target data and ★ indicates training with multi-target data.

Table 7: Comparison of SSFT with baselines on three math reasoning tasks, where all methods use Llama3.1-8B-Instruct as the base model and are trained on the `s1k-4mixed-reasoning`. SFT-mixed-distill denotes standard SFT where different traces for the same prompt are treated as independent training examples. SSFT (random $\sigma$) denotes SSFT with a random bipartite matching. For Pass@1, SSFT uses a graph-based heuristic: it prompts the forking token, `<think i>`, with the largest number of connected edges in the learned matchings; as a result, `<think 4>` is selected for SSFT-Llama3.1-8B and `<think 1>` for SSFT-Llama3.1-8B (random $\sigma$).

| | AIME 2024 | | AIME 2025 | | MATH-500 | |
|---|---|---|---|---|---|---|
| Model | Pass@1 | Cons@32 | Pass@1 | Cons@32 | Pass@1 | Cons@32 |
| Llama3.1-8B-Instruct | 3.65 | 13.33 | 0.94 | 6.67 | 47.53 | 65.40 |
| SFT-mixed-distill-Llama3.1-8B | 4.90 | 13.33 | 4.90 | 10.00 | 61.95 | **80.00** |
| SSFT-Llama3.1-8B (random $\sigma$) | 6.77 | 13.33 | 4.79 | 10.00 | **62.22** | 79.60 |
| SSFT-Llama3.1-8B | **6.98** | **16.66** | **5.79** | **20.00** | 62.04 | **80.00** |

△ indicates training with single-target data and ★ indicates training with multi-target data.

## A.7 EXTENDED RELATED WORK.

Aggarwal & Welleck (2025); Zhang et al. (2025) are seminal works that use RL to learn a policy controlling how long a reasoning model thinks. During RLVR, Aggarwal & Welleck (2025) appends an explicit "Think for $n$ tokens" instruction to the input prompt and adds a length penalty based on the exact deviation. Zhang et al. (2025) instead appends a higher-level length description and adjusts the length penalty accordingly. We list the differences from our method below, though we believe our method is complementary to many of these RL works, since RLVR requires a strong base model (Yue et al., 2025), which SSFT aims to provide.

**Our focus on diversity and correctness vs. their focus on efficiency.** They first enforce a relatively small token budget, $\leq 4096$, and then study accuracy. Our work instead aims to address the plateaued or lower performance caused by overthinking. Even if one particular reasoning mode does not improve with longer CoT, other reasoning modes can provide distinct and still potentially

long CoT. We use the distribution of thinking counts as an unbiased metric, in addition to accuracy differences, to assess diversity. We think length is a good diversity indicator for us because SSFT does not explicitly impose any length regularization. Controlling reasoning length through global forking tokens is an emergent behavior, in contrast to these RL works that use explicit length penalties in their objectives.

**Experiment settings.** Our SFT-OpenR1-93k-7B is a strong enough baseline scoring $46.15\%$ on AIME24, compared to $26\%$ by L1-Exact-7B (Aggarwal & Welleck, 2025), and $30\%$ by TLDR-7B-4k (Aggarwal & Welleck, 2025). Part of the difference likely comes from setup asymmetries: our baseline does not constrain the token limit and uses teacher distillations, whereas these works rely on DeepSeek-R1-Distill-7B, a base model that has gone through SFT using private data. So we would need to run a non-trivial number of experiments to first verify whether their length-controlled optimization remains competitive at much larger token budgets (at least 15,000), and whether their results hold with a base SFT model fine-tuned with public data. Even then, the research question would be whether their RL methods can make SSFT unnecessary, so we would need to compare SSFT plus their method against SFT plus their method. We plan to investigate the synergy between an SSFT base model and different RL methods in future work.

## A.8 ALGORITHM: SSFT IMPLEMENTATION

Algorithm 1 presents the core SSFT implementation with optimal bipartite matching. **In practice, the nested-loop computation used to populate $C$ is fully vectorized and can be executed in a single forward pass**. This does not blow up VRAM because (i) we do not store activations for these cost evaluations (no backprop through matching costs), and (ii) We only need to use the first $T_L < T_r$ NTP losses to compute the matching cost, as the NTP loss over the first few thousand tokens can already differentiate many reasoning traces in terms of their modes. Nevertheless, our code also supports matching over the full $T_r$ tokens by chunking the computation into a few batches, so this step does not become a VRAM bottleneck. Fine-tuning on 1k questions with 4 traces each, SSFT (optimal matching) took 6.5 h for 6 epochs, compared to 6.1 h for standard SFT, adding only a small overhead.

To support training with variable-size parallel generations, we propose a simple and scalable implementation that incurs no additional VRAM overhead. Rather than concatenating diverse reasoning traces along the sequence dimension, which would require complex sequence parallelism for memory efficiency, our algorithm expands variable-size parallel generations along the batch dimension under distributed training (Appendix A.8).

The primary VRAM bottleneck in SSFT remains the backpropagation Step 8, regardless of whether we use optimal or random matching, because the effective batch size scales with $M$. To mitigate this, we split the backward pass into several gradient-accumulation steps. Although our experiments use the same number of reasoning traces per question, we also support variable number of targets using our queue-based batching in Algorithm 2. The complication arises when using distributed training with a variable-sized batch, as different processes require the same per-device batch size to perform collective operations. This is mitigated by padding with PAD" sequences to align batch sizes. Our implementation minimizes the number of "PAD" sequences by storing a variable number of targets in a queue and dequeuing multiple items to form a per-device global batch, so smaller batches can be stitched together instead of always being padded.

## A.9 TRAINING DETAILS

### A.9.1 TRAINING DATASETS

**32B experiments with questions from s1(Main):** We explain the process of generating our training dataset for experiments in Table 1, Figure 3, Figure 4, Figure 5, Figure 6. First, we use the 1000 questions from s1 (Muennighoff et al., 2025) and populate a pool of reasoning traces by distilling from GPT-OSS-120B-high reasoning, GPT-OSS-120B-medium reasoning, DeepSeek R1, Gemini Flash2.0 Thinking, and Claude4/4.1. We use temperature 1.0, maximum length of 32768, and sets high reasoning effort unless specified. We generate two traces per teacher model. We use Claude3.5

---

**Algorithm 1** Set Supervised Fine-tuning (SSFT)

---

**Require:** • $\pi_{\boldsymbol{\theta}}$: base model
    • N: Number of global forking tokens $\{g^{(i)}\}_{i=1}^{N}$
    • $\mathcal{D}$: Dataset with (at most) M reasoning traces per question
    • B: Global batch size
    • $T_L$: The first $T_L$ number of tokens to match in matching cost (Equation 5).
**Ensure:** Output $\pi_{\boldsymbol{\theta}}$
 1: **for** each training step **do**
 2:  **for** $k = 1, ..., B$ **do**
 3:   Sample an input prompt and the corresponding reasoning traces $(\mathbf{x}_k, \{\mathbf{r}_k^{(j)}\}_{j=1}^{M}) \sim \mathcal{D}$
 4:   Initialize cost matrix $C \in \mathbb{R}^{N \times M}$
 5:   **for** $i = 1, ..., N$ **do**
 6:    **for** $j = 1, ..., M$ **do**
 7:     Compute the matching cost between $g^{(i)}$ and $\mathbf{r}_k^{(j)}$ by Equation 5.

$$\mathcal{L}_{\text{matching}}\left(g^{(i)}, \mathbf{r}_k^{(j)}\right) = -\text{sg}\left(\frac{1}{T_L}\sum_{t=1}^{T_L} \log \pi_{\boldsymbol{\theta}}\left(\mathrm{r}_{k,t}^{(j)}|\mathbf{x}_k, g^{(i)}, \mathbf{r}_{k,<t}^{(j)}\right)\right) \tag{5}$$

 8:     Store the matching cost in C.

$$C(i, j) = \mathcal{L}_{\text{matching}}\left(g^{(i)}, \mathbf{r}_k^{(j)}\right) \tag{6}$$

 9:    **end for**
10:   **end for**
11:   Compute optimal matching $\hat{\boldsymbol{\sigma}}_k$ between $\{g^{(i)}\}_{i=1}^{N}$ and $\{\mathbf{r}_k^{(j)}\}_{j=1}^{M}$. Hungarian algorithm (Kuhn, 1955) can be applied to C to efficiently compute Equation 7 (Equation 1).

$$\hat{\boldsymbol{\sigma}}_k = \arg\min_{\boldsymbol{\sigma} \in \mathfrak{S}_P} \sum_{j=1}^{M} C(\boldsymbol{\sigma}(j), j) \tag{7}$$

12:  **end for**
13:  Compute the empirical *set language modeling loss* (Equation 8):

$$\mathcal{L}_{\text{Hungarian}}(\boldsymbol{\theta}) = -\frac{1}{B}\sum_{k=1}^{B}\left[\sum_{j=1}^{M}\sum_{t=1}^{T_\mathbf{r}} \log \pi_{\boldsymbol{\theta}}\left(\mathrm{r}_{k,t}^{(j)}|\mathbf{x}_k, g^{(\hat{\boldsymbol{\sigma}}_k(j))}, \mathbf{r}_{k,<t}^{(j)}\right)\right] \tag{8}$$

14:  Update model parameters $\boldsymbol{\theta}$ using gradients $\nabla_{\boldsymbol{\theta}}\mathcal{L}_{\text{Hungarian}}(\boldsymbol{\theta})$
15: **end for**

---

---

**Algorithm 2** Queue-based Distributed SSFT with variable number of traces for each question

---

**Require:** • $\pi_{\boldsymbol{\theta}}$: base model
  • N: Number of global forking tokens $\{g^{(i)}\}_{i=1}^{N}$
  • $\mathcal{D}$: Dataset with (at most) M reasoning traces per question, variable $m$ number of traces per question
  • B: Global batch size
  • $T_L$: The first $T_L$ number of tokens to match in matching cost (Equation 5).
  • $b$: Original per-device global batch size ($b \geq$ M), This is "micro batch size*original grad accumulation steps"

**Ensure:** Output $\pi_{\boldsymbol{\theta}}$

1: **for** each epoch **do**
2:     Initialize Queue $Q$ for storing a sequence of $(\mathbf{x}_k, \{\mathbf{r}_k^{(j)}\}_{j=1}^{m})$ where $m$ is a variable number that differs between input questions and different processes (GPUs)
3:     Initialize Queue $q$ for storing a sequence of sizes of sets $m$.
4:     **for** every $(\mathbf{x}_k, \{\mathbf{r}_k^{(j)}\}_{j=1}^{m}) \in \mathcal{D}$ **do**
5:         $Q \leftarrow Q.enqueue((\mathbf{x}_k, \{\mathbf{r}_k^{(j)}\}_{j=1}^{m}))$
6:         $q \leftarrow q.enqueue(m)$
7:         $all\_q\_list = All\text{-}gather(q)$
8:         Initialize list $temp\_batch$
9:         **while** All processes have at least $b$ sequences based on $all\_q\_list$ **do**
10:            **while** $temp\_batch$ does not have at least $b$ sequences **do**
11:                $temp\_batch \leftarrow temp\_batch.append(Q.dequeue())$
12:                q.deque()
13:            **end while**
14:            compute the the maximum per_device global batch size $b_{max}$ currently in all processes using $all\_q\_list$ (inferred, no collective operation)
15:            Pad $temp\_batch$ to size $b_{max}$ by appending "pad sequences" as needed.
16:            Update all entries in $all\_q\_list$ based on inferred usage
17:            Perform one SSFT training step, SSFT ($\pi_{\boldsymbol{\theta}}, temp\_batch, b_{max}$)
18:        **end while**
19:    **end for**
20: **end for**

---

Sonnet to extract the answer from the distilled solutions and compare with the ground-truth answer. The correctness of these distilled traces are shown in 8.

**For s1k-4mixed-reasoning dataset**, we sample 4 traces per question from this pool, so the dataset consists of 1000 questions, each paired with 4 reasoning traces. This dataset was used to fine-tune **SSFT-32B**, **SSFT-32B (random $\sigma$)**, and **SFT-mixed-distill-32B**.

For fine-tuning **SFT-OSS-distill-32B** model, we only use the 1000 traces from "Run1" GPT-OSS-120B with high reasoning effort.

For obtaining the visualizations in Figure 6 and Figure 8, we fine-tune models using the same teacher models for all 1000 questions. we always choose the 4 traces from "Run1" of GPT-OSS-120B-high, GPT-OSS-120B-medium, DeepSeek R1, and Gemini Flash2.0 Thinking. As mentioned in Section 3.3 and Figure 6, we fine-tune under 3 bipartite matching hyperparameters to conduct this case study.

|  | GPT-OSS-120B-high | GPT-OSS-120B-medium | DeepSeekR1 | Gemini Flash2.0 Thinking | Claude Opus4/4.1 |
|---|---|---|---|---|---|
| Run1 | 796/1000 | 769/1000 | 620/1000 | 538/1000 | 656/1000 |
| Run2 | 785/1000 | 753/1000 | 641/1000 | 545/1000 | 647/1000 |

Table 8: The number of correct reasoning traces distilled for the 1,000 questions in s1 by different teacher models. This evaluation is done by Sonnet comparing the predictions and the ground-truth answers. We see that GPT-OSS has the highest accuracy for s1 dataset.

### A.9.2 SSFT TRAINING DETAILS

For consistency, we use Qwen2.5-32B-Instruct (Yang et al., 2025a) as the base model for all of our 32B experiments. We use standard fine-tuning hyperparameters: we train for 6 epochs with a global batch size of 32, which is derived from 4 gradient accumulation steps and distributed training with 8 GPUS ($4 \times 8 = 32$). This results in 756 gradient steps. The maximum sequence length is set to 32,768. We train with bfloat16 with a learning rate of $1e-5$ warmed up linearly for 5% and then decayed to 0 using a cosine schedule. We choose AdamW optimizer (Loshchilov & Hutter, 2017) with $\beta_1 = 0.9$, $\beta_2 = 0.95$, and weight decay $1e-4$. We only backpropagates the completion loss, which is the loss on reasoning traces and the answers. Fine-tuning **SSFT-32B** plus loggings took 6.5 hours on 8 NVIDIA B200 GPUs using PyTorch FSDP, Liger Kernel (Hsu et al., 2024) for fused cross entropy loss, and FlashAttention-2 (Dao, 2023) for fused attention computation. Fine-tuning **SSFT-32B (random $\sigma$)** took 6.3 hours, and Fine-tuning **SFT-mixed-distill-32B** took 6.1 hours. Even our baseline **SFT-OSS-distill-32B** with only one trace per question, and our attempt to reproduce s1.1 took 1.66 hours, which is longer than the time reported by Muennighoff et al. (2025). This is due to using 8 GPUs instead of 16 GPUs, hardware and package differences. When training with s1k-4mixed-reasoning, we added one extra epoch from 5 epochs to 6 epochs, since we have 4x reasoning traces, but we did not linearly increase the number of epochs, as these traces can be similar, and the number of distinct questions is still 1,000. Overall, we made sure all of our models are fine-tuned with consistent hyperapameters.

### A.9.3 VISUALIZATION OF SSFT TRAINING DYNAMICS

Figure 9 shows the standard training dynamics of SSFT with optimal bipartite matching. The resulting model is SSFT-32B. The loss plotted here is Equation 3.

Figure 10 shows the evolution of bipartite matching during SSFT. Figures 10a and 10b show that the gap between optimal bipartite matching cost and non-optimal bipartite matching cost under other $\sigma$ keeps widening during training. This means that these reasoning traces are indeed starting to match unique global forking tokens. Even though SSFT effectively optimizes a non-stationary objective which depends on model parameters $\theta$, the widening gap shows the inner discrete optimization is converging as training goes. Figure 10c also confirms that the model learned some unique correlations between $\{\mathbf{g}^{(i)}\}_{i=1}^{6}$ and $\{\mathbf{r}^{(j)}\}_{j=1}^{4}$, as only a subset of matchings are still computed as optimal.

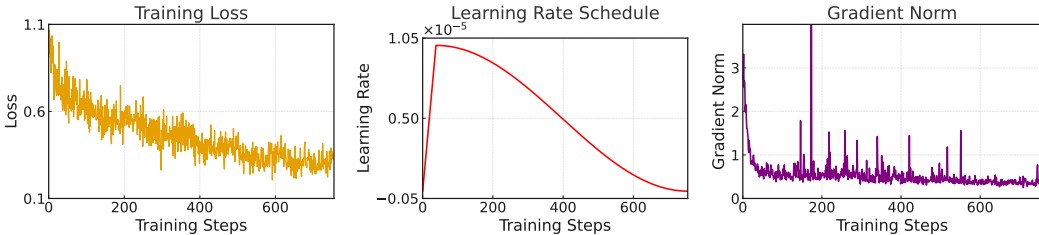

Figure 9: Training dynamics of SSFT-32B on s1k-4mixed-reasoning

Compared to Figure 6, we see more $\sigma$ accumulating mass towards the end. This is due to having more mixed diverse reasoning traces, so the model learned more intricate associations between these global forking tokens and truly diverse reasoning traces.

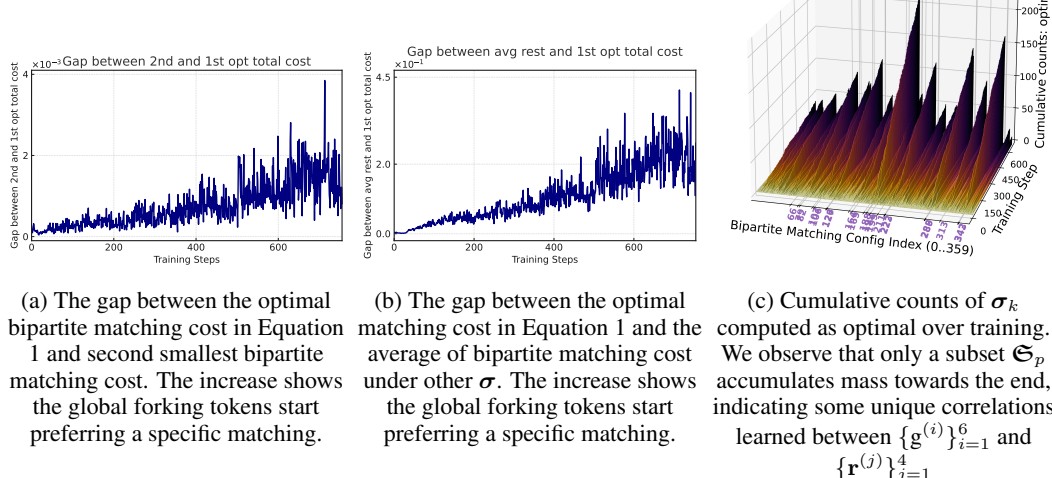

(a) The gap between the optimal bipartite matching cost in Equation 1 and second smallest bipartite matching cost. The increase shows the global forking tokens start preferring a specific matching.

(b) The gap between the optimal matching cost in Equation 1 and the average of bipartite matching cost under other $\sigma$. The increase shows the global forking tokens start preferring a specific matching.

(c) Cumulative counts of $\sigma_k$ computed as optimal over training. We observe that only a subset $\mathfrak{S}_p$ accumulates mass towards the end, indicating some unique correlations learned between $\{\mathbf{g}^{(i)}\}_{i=1}^6$ and $\{\mathbf{r}^{(j)}\}_{j=1}^4$

Figure 10: Dynamics of bipartite matching during the fine-tuning of SSFT-32B on s1k-4mixed-reasoning.

### A.9.4 ABLATION STUDY TRAINING DETAILS (REMOVING HIGH QUALITY SMALL DATASET)

For this ablation study, we choose Open-R1-Math220k default split, which has 93,000 math questions and $2 \sim 4$ traces. Since Qwen2.5-Math-7B is a widely fine-tuned model using this dataset, we also choose it as our base model. We train for 3 epochs using 8 A100 GPUs, which took around 4 days. Our hyperparameters are mostly consistent with the recommended hyperparameters by Face (2025). We fine-tune both SSFT-OpenR1-93k-7B and SFT-OpenR1-93k-7B with a maximum length of 32768, learning rate of $4.0e - 05$ warmed up linearly for $3\%$ and decayed to 0 following cosine schedule, 8 gradient accumulation steps. For our SSFT method, we reserve N = 4 global forking tokens, and use the first 1000 tokens for matching. Again, only the completion loss is used for optimizing the model parameters.

### A.9.5 OTHER TRAINING SETUP AND HYPERPARAMETERS

Table 9 shows the hyperparameters used for all of our key experiments.

Table 9: Trainning details of SSFT and GFPO. SFT-mixed-distill-32B-tags-GRPO optimizes over full sequences, whereas GFPO only optimizes `<think i>`, tokenized in 4 tokens. 8xB200 is used for full GRPO.

| SSFT | | GFPO (GRPO on Forking Tokens) | |
|---|---|---|---|
| **Parameter** | **Value** | **Parameter** | **Value** |
| Global Batch Size | 32 | Global Batch Size | 128 |
| GPUs | 8xB200 | GPUs | 8xA100 |
| Max Sequence Length | 32768 | Max Generation Length | 31000 |
| N Global Forking Tokens | 6 | Max Prompt Length | 1024 |
| M distillation targets | 4 | Number of rollouts | 5 |
| L Matching Length | 1000 | PPO minibatch size | 128 |
| Learning Rate | 1e-5 | Learning Rate | 1e-6 |
| Adam $\beta_1$ | 0.9 | Adam $\beta_1$ | 0.9 |
| Adam $\beta_2$ | 0.95 | Adam $\beta_2$ | 0.95 |
| Warmup ratio | 5% | Rollout Temperature | 0.7 |
| Epochs | 6 | Epochs | 1 |
| | | KL coeffcient | 0.001 |
| **Training data for math models** | | | |
| `s1k-4mixed-reasoning` s1 questions with 4 mixed traces each. | | 1280 questions from `DAPO-Math-17k` | |
| **Training data for coding models** | | | |
| `code1k-4mixed-reasoning` - 1k questions from Open-Thoughts - 4 R1 traces each | | 1280 questions from `Intellect-2-RL` | |
| **Base models being used** | | | |
| Qwen2.5-32B-Instruct Qwen3-4B-Base Llama3.1-8B-Instruct | | SSFT-32B SFT-mixed-distill-32B-tags SSFT-32B (random $\sigma$) SSFT-32B-code (before RL) SFT-mixed-distill-32B-code (before RL) SSFT-32B (random $\sigma$)-code (before RL) | |
| Qwen2.5-Math-7B-Instruct. Separate training details and hyperparameters reported in Appendix A.9.4 | | | |

## A.10 EXAMPLES OF SSFT-32B REASONING TRACES INITIATED BY DIFFERENT GLOBAL FORKING TOKENS

We randomly picked a set of six parallel generations from those where SSFT-32B correctly answered a problem from AIME 2025 under all six global forking tokens. This is question 11.

---

**Problem 11 (AIME25)**

A piecewise linear periodic function is defined by

$$f(x) = \begin{cases} x, & \text{if } x \in [-1, 1), \\ 2 - x, & \text{if } x \in [1, 3), \end{cases} \qquad \text{and} \qquad f(x + 4) = f(x) \text{ for all real } x.$$

The graph of $f(x)$ has a sawtooth pattern. The parabola $x = 34y^2$ intersects the graph of $f(x)$ at finitely many points. The sum of the $y$-coordinates of these intersection points can be expressed in the form

$$\frac{a + b\sqrt{c}}{d},$$

where $a, b, c, d$ are positive integers, $\gcd(a, b, d) = 1$, and $c$ is squarefree. *Find* $a + b + c + d$.

---

Here are the 6 parallel generations by SSFT-32B. We see distinct reasoning traces produced by prompting with a unique $g^{(i)} \in \{g^{(i)}\}_{i=1}^6$. The differences are highlighted in red comments.

<think 1> — Periodization + Quadratic Families + Vieta/Telescoping

```
<|im_start|>assistant
<|im_start|>
```
<think1>

"write $x = t + 4k$ with $t \in [-1, 3)$" ... ... "Case 1 ($t \in [-1, 1)$, $y = t$) ... Case 2 ($t \in [1, 3)$, $y = 2 - t$)" ... ... $34t^2 - t - 4k = 0$ and $t = \dfrac{1 \pm \sqrt{1 + 544k}}{68}$ ... ... "let $u = 2 - t$" and $u = \dfrac{-1 \pm \sqrt{273 + 544k}}{68}$ ... ... *Thus* $u = \dfrac{-1 + 5\sqrt{185}}{68}$. ... ... "telescopes to rational constant $9/34$" ... ... *([Comment:]* periodize $x = t + 4k$, solve two quadratic families, use Vieta pair-sums; edge-period gives the irrational $y$)

</think1>
```
<|im_start|>answer
```
Answer: .... $\boxed{259}$

<think 2> — Nearest-Multiple-of-4 Reduction

```
<|im_start|>assistant
<|im_start|>
```
<think2>

Start from $y = f(34y^2)$ ... ... write $34y^2 = 4k + r$ with $r \in [-1, 3)$ (choose $k = \lfloor \frac{34y^2 + 1}{4} \rfloor$) ... ... Branch A ($r \in [-1, 1)$): $y = r \Rightarrow 34y^2 - y - 4k = 0$ ... ... Branch B ($r \in [1, 3)$): $y = 2 - r \Rightarrow 34y^2 + y - (2 + 4k) = 0$ ... ... Vieta: pair-sums $+\frac{1}{34}$ and $-\frac{1}{34}$ cancel over full periods ... ... only the edge/partial period remains ($k = 8$, descending branch) ... ... $y = \dfrac{-1 + \sqrt{273 + 544 \cdot 8}}{68} = \dfrac{-1 + 5\sqrt{185}}{68}$ ... ...

*([Comment:]* reduce to $r \in [-1, 3)$ via nearest multiple of 4; full periods cancel by $\pm\frac{1}{34}$ pair-sums; only the boundary $k = 8$ root contributes)

</think2>
```
<|im_start|>answer
```
Answer: .... $\boxed{259}$

<think 3> — Paper-Style Sections; Explicit Pair-Sums and Totals

```
<|im_start|>assistant
<|im_start|>
```
<think3>

"Now we need sum of all $y$-coordinates" ... ... $\displaystyle\sum_{n=0}^{8} y_{1,n} + \sum_{n=0}^{7} y_{C,n}$ ... ... $y = \dfrac{-1 + \sqrt{4625}}{68}$

... ... "Therefore $S_2 = \dfrac{5\sqrt{185} - 17}{68}$," ... ... "Total sum $S$ ..." ... ... *([Comment:]* formal write-up; separates sums by family, explicitly displays the boundary $y = \frac{-1+\sqrt{4625}}{68}$, then totals)

</think3>
```
<|im_start|>answer
```
Answer: .... $\boxed{259}$

<think 4> — Set/System Framing Before Algebra

```
<|im_start|>assistant
<|im_start|>
```
<think4>

"the set $\{(x, f(x))\}$ ... Intersecting it with the parabola $x = 34y^2$ requires solving $y = f(x)$, $x = 34y^2$." ... ... (then proceeds to the same two-case algebra) ... ...
$u_+ = \dfrac{-1 + \sqrt{273 + 544 \cdot 8}}{68} = \dfrac{-1 + 5\sqrt{185}}{68}$. ... ... (*[Comment:]* definition-first framing; after setting up the system, it reaches the same boundary $u_+$ value on the partial period)
</think4>
`<|im_start|>`answer
Answer: .... $\boxed{259}$

---

`<think 5>` — Outline & Symmetry: Full Periods Cancel, Last Period Contributes

`<|im_start|>`assistant
`<|im_start|>`
<think5>
"the sum is zero ... That's a key insight." ... ... "for each full period ($k = 0$ to $k = 7$), the sum ... is zero ... now consider the partial period $k = 8$" ... ... *Compute* $u_{pos}(k=8) = \dfrac{-1 + \sqrt{4625}}{68}$. ... ... (term-by-term cancellation examples for $k = 0, 1$) ... ... (*[Comment:]* symmetry explains full-period cancellation; the lone boundary contribution at $k = 8$ supplies the irrational term)
</think5>
`<|im_start|>`answer
Answer: .... $\boxed{259}$

---

`<think 6>` — Stream-of-Consciousness Scratchwork

`<|im_start|>`assistant
`<|im_start|>`
<think6>
"Okay, so I need to solve this problem ... Let me try to break this down step by step." ... ... "Wait, actually, no. Wait," ... ... $y_{A,8} = \dfrac{-1 + \sqrt{4625}}{68}$. ... ... (*[Comment:]* exploratory self-checks; reaches the same boundary $y$ on the partial period)
</think6>
`<|im_start|>`answer
Answer: .... $\boxed{259}$

## A.11 ALTERNATIVE IMPLEMENTATION OF TRAINING WITH MULTI-TARGET SEQUENCES

Figure 11 shows an alternative implementation for training with multiple target parallel sequences. Instead of flattening multi-target data along the batch dimension, we can also concatenate parallel reasoning traces along the sequence dimension, modify the causal attention matrix, and position ids for positional embeddings. While easy to implement, this version extends the sequence length and cannot perform gradient accumulation along the batch dimension to maintain the same VRAM as training with a single reasoning trace. Since most fine-tuning already sets the micro-batch size to 1 due to the increased number of reasoning tokens already present in a single reasoning path, this implementation also cannot further shrink the micro-batch size to accommodate the extended context length. In addition, the code that uses the initial flash-attention implementation, which requires causal attention matrix, is not compatible with this setup. In terms of choosing the number of sequences in a global batch size, this implementation is also less flexible.

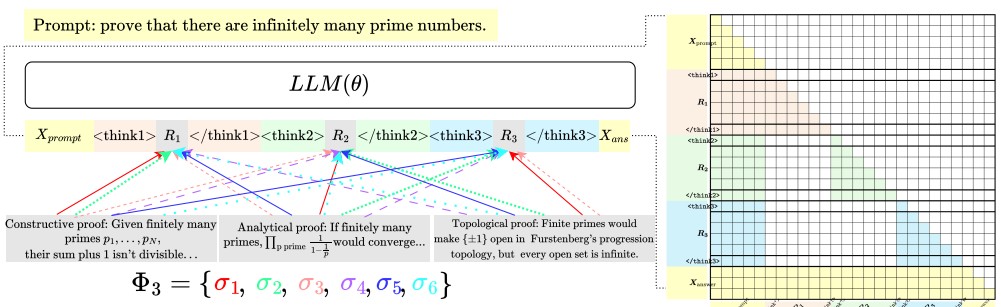

Figure 11: An illustration of training with multiple parallel targets along the sequence dimension.

