# OpenReview forum: "Training Large Language Models To Reason In Parallel With Global Forking Tokens"
_ICLR.cc/2026/Conference — ICLR 2026 Poster_

### Official Review · Reviewer_8WvB · 2025-10-28

**Soundness:** 3
**Presentation:** 3
**Contribution:** 3
**Rating:** 6
**Confidence:** 3

**Summary:**

This paper tackles "mode collapse" in LLMs trained on diverse reasoning traces. It introduces Set Supervised Fine-Tuning (SSFT), which frames parallel reasoning as a set prediction problem. SSFT uses a set of global forking tokens $\{g^{(i)}\}$ and multiple reasoning traces $\{r^{(j)}\}$. At each step, it finds an optimal bipartite matching $\hat{\sigma}$ between tokens and traces by minimizing the NTP loss. The model only optimizes this set-based loss, $\mathcal{L}_{Hungarian}(\theta)$. This forces tokens to learn distinct reasoning modes, improving both Pass@1 (single) and Cons@k (voting) accuracy on reasoning benchmarks.

**Strengths:**

1. This paper introduces a novel problem formulation, treating parallel reasoning as a "set-of-next-token-prediction" task. It applies a set-based loss via bipartite matching (inspired by DETR in vision) to LLM finetuning.
2. The empirical evaluation is rigorous. By comparing SSFT against baselines trained on the exact same data (SFT-mixed) and an ablation (SSFT-random), the paper proves the method's core components are responsible for its success. Visualizations (Fig. 4 vs 5) powerfully confirm that SSFT prevents the "mode collapse" that plagues naive finetuning.
3. The paper is clearly written, and its core, complex mechanism is made simple and intuitive by the visualization in Figure 1, which illustrates the entire training step.
4. This work provides the first effective method for multi-teacher distillation, allowing a single model to learn diverse reasoning strategies. It significantly improves the sample efficiency of parallel decoding (Cons@k) and introduces "global forking tokens" as a new, powerful form of steerable generation.

**Weaknesses:**

1. Computational Scalability: The SSFT algorithm requires $\mathcal{O}(N \times M)$ forward passes (where $N$=tokens, $M$=traces) at each training step to build the cost matrix. This is a potential bottleneck that may not scale efficiently if researchers want to use many more tokens or traces.
2. Reliance on Matching Heuristic: The optimal matching is computed using only the first $T_L=1000$ tokens of each trace for efficiency. The paper lacks an ablation study on this critical $T_L$ hyperparameter, making it unclear if the method is robust for reasoning paths that only diverge after this prefix.
3. Limited Task Scope: The experiments are confined to mathematical and structured-knowledge reasoning. It is not demonstrated how SSFT would perform on more open-ended, creative, or argumentative tasks where "correctness" and "diversity" are less clearly defined.
4. Weak Justification for Pass@1 Token: The heuristic for choosing the best token for Pass@1 inference (the one with the "largest coverage") is not well-justified. The paper does not provide strong evidence for why this "flexibility" heuristic should correlate with the highest single-attempt accuracy.

**Questions:**

1. The caption of the table should be placed above.
2. See Weakness.

---

> ### Author Response · Authors · 2025-11-25
> **Author Response #1 to Reviewer 8WvB**
>
> We sincerely thank you for recognizing the novelty of our set-prediction formulation and for highlighting the rigor of our evaluations. We also greatly appreciate your praise of our visualizations (Figures 1, 4, and 5), which we view as the most representative figures of the paper’s core contributions. New experimental results are summarized in the **``Summary of Concerns and New Experimental Results’’ section of the general response** and are also presented in a self-contained form in the **appendix of the latest revision.** We thank you for your thoughtful feedback and have done our best to address the identified weaknesses (**W#**) and questions (**Q#**).
>
>
> ## **W1: Computational complexity**
> Although computing all $MN$ matching costs at full generation length can be expensive, we clarify in **Section 2.2**. that using a smaller matching length **L** can reduce such complexity.  Our ablation in **Figure 7 (Section 4)** shows that the $L=1000$ setting used throughout our experiments yields a significant performance gain while adding only $\frac{6.5h−6.3h}{6.3h}=3.2$\% training time in our setup. When additional compute time permits, $L$ can be increased further and stopped once performance plateaus.
>
>
> ## **W2: Lacking an ablation study on matching length L**
> This is partially addressed in our response to **W1**. **Figure 7** shows that SSFT with various matching lengths $L$ (100, 1000, 2500, 3500) consistently outperforms the non-optimal matching baseline, and performance already plateaus at $L=1000$ in our setting. For practitioners, a reasonable starting point is $\frac{\text{max seq len}}{MN}$ for maximum efficiency; $L$ can then be increased incrementally and stopped once performance saturates.
>
> ## **W3: Limited Task Scope**
> We fine-tuned on only 1K code-generation questions (each with four traces) for the SSFT stage and 1.28K questions for the GFPO stage, and evaluated Pass@1 on LiveCodeBench (LCB). Full results, along with baselines, are added to **Table 2 (Section 4) and Table 5 (Appendix A.2)**. SSFT instills diverse, specialized reasoning modes **without collapse**, and RL can quickly leverage them to identify an optimal forking path and reasoning mode for each coding task, thereby significantly improving Pass@1. SSFT achieves 52.07% on LCBv5, which is 4.94% higher than the strongest baseline and 28.72% higher than the base model. SSFT coding model also shows better OOD generalization to math reasoning tasks.
>
> ## **W4: Weak justification for Pass@1 heuristic**
> Besides including more clarifications for our heuristic in **Appendix A.1 (Background)**, we apply a very small amount of RL to the distribution of global forking tokens. In **Sections 2.3 and 2.4**, we explain this Global Forking Policy Optimization (GFPO) and the new Pass@1 inference based on the distributions of forking tokens. We do not propose a new RL algorithm; rather, global forking tokens make additional RL extremely simple for learning a distribution that eliminates the need for a Pass@1 heuristic.
>
> Results for Pass@1 without the heuristic are denoted by “-GFPO” in **Table 1 (Section 3)** and **Table 4 (Appendix A.1)**. We also applied RL to the other baselines to ensure the comparisons remain fair throughout. The Cons@6 and Cons@32 results for SSFT-32B-GFPO are reported in **Table 1 (Section 3)**, showing that this additional RL step does not reduce performance under parallel test-time compute; in fact, it improves Cons@6 on AIME25 to 78.48%.
>
>
> ## **Q1: Caption of the table should be placed above.**
> Thank you for pointing this out. In the latest revision, we have moved all table captions to appear above their respective tables.

---

> > ### Comment · Reviewer_8WvB · 2025-11-27
> >
> > Thanks for your rebuttal, I raised my soundness score: 3 -> 4

---

### Official Review · Reviewer_Ut4D · 2025-10-29

**Soundness:** 3
**Presentation:** 3
**Contribution:** 3
**Rating:** 6
**Confidence:** 2

**Summary:**

The paper treats parallel reasoning as a set-of-next-token-prediction problems. The paper proposes Set Supervised Fine-Turning (SSFT). They incorporate a set-based global loss and use the loss for Supervised Fine-Turning (SFT). Experiments on multiple reasoning benchmarks show our SSFT method consistently outperforms SFT.

**Strengths:**

1. I think the paper treats parallel reasoning as a set-of-next-token-prediction problem is an interesting and useful idea.
2. The paper focuses on an important question.
3. The motivation and intuition under the paper is pretty clear and easy to follow.

**Weaknesses:**

1. The experiments are only conducted with a single base model. But I think it will be helpful is models with different size and from different model families can also be used.
2. I think the comparison to baseline is a bit unfair. For (MultiTarget), they use one <think> token for all four traces and treating them as four individual data points. This can make all traces mixed together and confuse the model. I think a fairer comparison is to use different tokens for different traces. As you do for SSFT. (Single-Target △) models trained with one trace per question and (Multi-Target ⋆) models trained with four traces per question. Does it mean that the total number of training tokens is different?
3. It's unclear to me how robust the algo is, especially the matching part. I think there will be probamatic if the generated trace of the base model is very similar to one if the teacher model. For example, if the base model is deepseek-distilled, and deepseek is one of the teacher model.

**Questions:**

1. "To find the optimal bipartite matching for each input prompt, we consider only the first 1,000 tokens when computing the matching cost in Equation 2 for computational efficiency." Why did you only choose the first 1,000 tokens? Is it enough for computing the matching cost? How accurate is it?
2. Have you ever compared the methods to the RL method? [1,2] can also generate different length of the outputs guided by instructions/tokens.
3. When generate Training Dataset, what to do if a teacher model can't generate correct traces?
 [1] Zhang, Xuechen, et al. "Making small language models efficient reasoners: Intervention, supervision, reinforcement." arXiv preprint arXiv:2505.07961 (2025).

[2] Aggarwal, Pranjal, and Sean Welleck. "L1: Controlling how long a reasoning model thinks with reinforcement learning." arXiv preprint arXiv:2503.04697 (2025).

---

> ### Author Response · Authors · 2025-11-25
> **Author Response #1 to Reviewer Ut4D**
>
> We sincerely thank you for finding our set-prediction formulation interesting and useful. We also appreciate your recognition that the paper tackles an important question and is easy to follow. New experimental results are summarized in the **``Summary of Concerns and New Experimental Results’’ section of the general response** and are also presented in a self-contained form in the **appendix of the latest revision.** We thank you for your thoughtful feedback and have done our best to address the identified weaknesses (**W#**) and questions (**Q#**).
>
> ## **W1: Limited evaluation across model families and sizes.**
> We added the same experiments using Qwen3-4B-Base, from the new Qwen3 family at the 4B scale; results are in **Table 6 (Appendix A.3)**. Our original manuscript includes Qwen2.5-32B-Instruct and Qwen2.5-Math-7B-Instruct. We plan to add Llama3.1-8B-Instruct in the next revision.  **Table 6** shows that SSFT remains robust on a smaller 4B model from the newer Qwen3 family.  However, we do observe a smaller gap than with the larger 32B and 7B models in **Tables 1 and 3**. We show Pass@1 improvements below, comparing SFT-mixed baselines to SSFT.
> | Pass@1 Acc | Improvement from SFT to SSFT using Qwen3-4B-Base | Improvement from SFT to SSFT using Qwen2.5-Math-7B-Instruct | Improvement from SFT to SSFT using Qwen2.5-32B-Instruct |
> |-----------|---------------------------------------------------|------------------------------------------------------------|---------------------------------------------------------|
> | AIME24    | 23.44 - 21.46 = 1.98                             | 51.25 - 46.15 = 5.1                                       | 64.06 - 58.23 = **5.83**                                    |
> | AIME25    | 23.13 - 21.98 = 1.15                             | 35.52 - 34.17 = 1.35                                      | 58.13 - 51.96 = **6.17**                                    |
> | MATH500   | 81.85 - 78.76 = 3.09                             | 89.74 - 86.62 = **3.12**                                      | 90.02 - 88.49 = 1.53                                    |
>
> **Final revision**: we additionally included Llama3.1-8B-Instruct model as a base model for SSFT and all baselines. The reported results are in **Table 7 (Appendix A.3)**
>
> ## **W2: A bit unfair that the SFT-mixed-distill baseline does not use distinct $\texttt{<think\ i>\,}$ tags.**
> We appreciated and followed your suggestion and appended distinct think tags to the four traces, sampled without replacement. We observe an improvement, though it remains weaker than SSFT. We denote this as “SFT-mixed-distill-32B-tags” in **Table 1** and in other new experiments.  The improvements are shown below:
> |                      | AIME24 | AIME25 | MATH-500 | GPQA-D |
> |----------------------|-------:|-------:|---------:|-------:|
> | **Pass@1**           |        |        |          |        |
> | SFT-mixed-distill-32B        | 55.73 | 51.56 | 88.36 | 57.50 |
> | SFT-mixed-distill-32B-*tags* | 58.23 | 51.96 | 88.49 | 59.96 |
> | **Cons@6**           |        |        |          |        |
> | SFT-mixed-distill-32B        | 72.42 | 70.91 | 92.10 | 57.32 |
> | SFT-mixed-distill-32B-*tags* | 73.94 | 70.00 | 95.88 | 58.75 |
> | **Cons@32**          |        |        |          |        |
> | SFT-mixed-distill-32B        | 80.00 | 73.33 | 96.20 | 60.61 |
> | SFT-mixed-distill-32B-*tags* | 76.67 | 76.67 | 96.20 | 58.59 |
>
> ## **W3: Potential problem if one of the traces is similar to the ones generated by the base model.**
> Because we use one-to-one matching, a trace that is similar to the base model does not interfere with matching other distinct traces to the remaining unallocated global forking tokens. Those other tokens become more correlated with the distinct traces. Moreover, if traces similar to the ones from the base model make up a larger fraction of the data pool, we can increase the number of global forking tokens to create more unallocated slots.

---

> ### Author Response · Authors · 2025-11-25
> **Author Response #2 to Reviewer Ut4D**
>
> ## **Q1: Why did you choose L=1000 for the matching cost? Is it enough? How accurate is it?**
> We chose it to maximize the efficiency of computing the cost matrix in a single forward pass, $\frac{\text{max seq len}}{MN} = \frac{32768}{4*6} \approx 1000$, as our effective batch size for computing all matching costs increases by $MN$, and we added an ablation in **Figure 7 (Section 4)**, “Impact of Matching Length”, showing that this was indeed sufficient. Practitioners can further increase $L$ with a bit more compute and stop once performance plateaus.
>
> ## **Q2: Potential comparison with RL methods that control the reasoning length [1,2]?**
> Thank you for pointing out these papers.  We added an extended related work section in **Appendix A.6** and will include them in the final revision.  We explain that these referenced works focus on improving efficiency and work with low token budget <4096, whereas we focus on accuracy and diversity in complex math (AIME) tasks, where you need a much larger budget to be competitive. There are also important differences in the experimental setup, such as their use of DeepSeek-R1-Distill-Qwen-7B as the base model, which has already undergone SFT on reasoning traces. SSFT aims to improve such a base model and is complement to these RL works.
>
> [1] Zhang, Xuechen, et al. "Making small language models efficient reasoners: Intervention, supervision, reinforcement." arXiv preprint arXiv:2505.07961 (2025).
>
> [2] Aggarwal, Pranjal, and Sean Welleck. "L1: Controlling how long a reasoning model thinks with reinforcement learning." arXiv preprint arXiv:2503.04697 (2025).
>
> ## **Q3: What to do if a teacher cannot generate correct traces?**
> Our original manuscript (now **Table 9 in Appendix A.11.1**) shows that our traces have 60 – 70+% answer accuracy for s1 questions, yet we still observe significant improvements over SFT on reasoning benchmarks because the traces provide valuable intermediate reasoning processes. S1 also noted that including traces with incorrect answers can still improve their distilled model.  As shown in our experiments, SSFT can leverage such traces, including those with incorrect final answers, more effectively than SFT by preventing reasoning mode collapse. If a teacher’s intermediate reasoning is entirely incorrect, this issue will affect all distillation methods.

---

### Official Review · Reviewer_6BW4 · 2025-10-30

**Soundness:** 3
**Presentation:** 3
**Contribution:** 3
**Rating:** 6
**Confidence:** 4

**Summary:**

This paper proposes Set Supervised Fine-Tuning (SSFT), which introduces global forking tokens prior to generating parallel reasoning traces and frames parallel reasoning as a set prediction problem. This method expands the diversity of generated traces and keeps the accuracy of the generation at the same time.

**Strengths:**

1. The motivation of this work is reasonable, which points out the shortcomings of the current parallel scaling methods and the trade-off between diversity and accuracy.
2. Based on the motivation in (1), the logic of the design of the framework and matching algorithm makes sense.
3. The experiment part shows the effectiveness of this method compared to other baselines in both mathematical and STEM reasoning tasks.
4. This work also introduces a way to deploy the method efficiently, which is a good engineering contribution, although this might not be the main focus of this paper.

**Weaknesses:**

1. Some obvious and important typos, such as Fine-tuning not Fine-turninig.
2. The main result section might need more diversity on model families and model sizes.
3. Maybe an analysis of the generalization ability of this method is needed.

**Questions:**

1. Could you please include an analysis of out-of-distribution performance after training?
2. Could you please include 1 or 2 more different sizes of models in the main experiment part?
3. Could you please include 1 or 2 other model families to show the generalization of this method, such as Qwen3, Llama, or Phi?

---

> ### Author Response · Authors · 2025-11-25
> **Author Response #1 to Reviewer 6BW4**
>
> We sincerely thank you for your positive assessment of the soundness of our framework, matching algorithm, and motivation. We are also glad that you recognized our engineering contribution in making multi-teacher training efficient. New experimental results are summarized in the **``Summary of Concerns and New Experimental Results’’ section of the general response** and are also presented in a self-contained form in the **appendix of the latest revision.** We thank you for your thoughtful feedback and have done our best to address the identified weaknesses (**W#**) and questions (**Q#**).
>
>
> ## **W1: Some obvious and important typos.**
> We have corrected the typos and will continue polishing the language in future revisions.
>
> ## **W2: Limited evaluation across model families and sizes.**
> We added the same experiments using Qwen3-4B-Base, from the new Qwen3 family at the 4B scale; results are in **Table 6 (Appendix A.3)**. Our original manuscript includes Qwen2.5-32B-Instruct and Qwen2.5-Math-7B-Instruct. We plan to add Llama3.1-8B-Instruct in the next revision.  **Table 6** shows that SSFT remains robust on a smaller 4B model from the newer Qwen3 family.  However, we do observe a smaller gap than with the larger 32B and 7B models in **Tables 1 and 3**. We show Pass@1 improvements below, comparing SFT-mixed baselines to SSFT.
> | Pass@1 Acc | Improvement from SFT to SSFT using Qwen3-4B-Base | Improvement from SFT to SSFT using Qwen2.5-Math-7B-Instruct | Improvement from SFT to SSFT using Qwen2.5-32B-Instruct |
> |-----------|---------------------------------------------------|------------------------------------------------------------|---------------------------------------------------------|
> | AIME24    | 23.44 - 21.46 = 1.98                             | 51.25 - 46.15 = 5.1                                       | 64.06 - 58.23 = **5.83**                                    |
> | AIME25    | 23.13 - 21.98 = 1.15                             | 35.52 - 34.17 = 1.35                                      | 58.13 - 51.96 = **6.17**                                    |
> | MATH500   | 81.85 - 78.76 = 3.09                             | 89.74 - 86.62 = **3.12**                                      | 90.02 - 88.49 = 1.53                                    |
>
> **Final revision**: we additionally included Llama3.1-8B-Instruct model as a base model for SSFT and all baselines. The reported results are in **Table 7 (Appendix A.3)**
>
> ## **W3: Require analysis of out-of-distribution (OOD) generalization to other domains.**
> We added results on OOD generalization to LiveCodeBench in **Table 1 and Table 4 (Section 3.2) for the 32B model**, and in **Table 6 (Appendix A.3) for the 4B model**.  These OOD tasks are marked with an asterisk (*). Notably, Table 1 and Table 4 show SSFT improves generalization to code generation by about 4% over the baselines and about 19% over the base model in the 32B setting.  Since we also have new models fine-tuned using only coding traces, we report their OOD generalization to math tasks in **Table 5 (Appendix A.2.)**. A summary of this analysis is provided in **Appendix A.5.**
>
> **An example of OOD generalization analysis (Pass@1)**
> | Qwen2.5-32B-Instruct            | LCB: 23.35                 | AIME25: 10.40                 |
> |---------------------------------|----------------------------|-------------------------------|
> | Method + Base model         | Math Fine-tuning → LCB(v5)* | Code Fine-tuning → AIME25*   |
> | SFT+RL  (Qwen2.5-32B-Instruct)      | 37.72                      | 24.17                         |
> | SSFT+RL (Qwen2.5-32B-Instruct)     | 42.10                      | 32.82                         |
>
> ## **Q1-Q3:**
> These are addressed in **W2**, **W3**.

---

### Official Review · Reviewer_MjCW · 2025-11-01

**Soundness:** 3
**Presentation:** 3
**Contribution:** 3
**Rating:** 6
**Confidence:** 4

**Summary:**

This paper introduces **Set Supervised Fine-Tuning (SSFT)**, a novel framework designed to make Large Language Models (LLMs) generate multiple, diverse, and correct reasoning paths simultaneously. SSFT reframes parallel reasoning as a "set-of-next-token-prediction" task. It employs a set of learnable **"global forking tokens"** and a **bipartite matching** loss function to associate these tokens with different reasoning solutions during training. This approach enables the model to produce varied reasoning patterns when prompted with different forking tokens, effectively overcoming the typical trade-off between diversity and accuracy. Experiments show that SSFT-trained models significantly outperform standard Supervised Fine-Tuning (SFT) on challenging reasoning benchmarks.

**Strengths:**

1.  **High Innovation**: The SSFT framework and the concept of "global forking tokens" are highly original. Formalizing parallel reasoning as a set prediction problem with bipartite matching is a significant and clever departure from standard fine-tuning paradigms, effectively mitigating "mode collapse."
2.  **Effective and Elegant Method**: The use of the Hungarian algorithm for optimal matching allows the model to end-to-end learn associations between forking tokens and different reasoning styles without explicit labels. This design is both elegant and highly effective.
3.  **Strong Empirical Results**: The SSFT model demonstrates superior performance on difficult reasoning benchmarks like AIME and MATH-500. The ablation studies are thorough, clearly proving the necessity of the optimal matching component and showing a better diversity-accuracy balance compared to baselines.
4.  **High-Quality Presentation**: The paper is well-written and logically structured. The visualizations provide clear, intuitive evidence for the method's effectiveness and interpretability.

**Weaknesses:**

1.  **Implementation Complexity**: SSFT's training process, which requires solving a bipartite matching problem at each step, is more complex than standard SFT. This could pose a barrier for application on larger-scale models or datasets.
2.  **Data Dependency**: The method's success hinges on access to a high-quality, diverse set of reasoning paths, which can be costly to obtain. The performance might degrade if such data is unavailable. But considering that this work explicitly aims to preserve the model’s diversity under such settings, this weakness is not particularly limiting.
3.  **Heuristic for Single-Path Inference**: The strategy for choosing a forking token in single-pass (Pass@1) inference is based on a heuristic (the token with the most connections). This choice lacks strong theoretical justification and could be explored further.

**Questions:**

- What is the expected performance of SSFT on much smaller (<7B) or larger (>70B) models?
- Regarding the number of global forking tokens, N, I would like to explore the potential impact of its fixed setting. The inherent diversity of reasoning paths differs for any given problem. Is it possible that a uniform N value could create redundancy for simpler problems while failing to fully capture the diversity of more complex ones? I am curious about the authors' perspective on the feasibility and potential of dynamically adjusting N for each problem. For instance, could performance be improved by a mechanism that predicts or assigns an optimal N' value tailored to each specific question?
- This paper brilliantly demonstrates how SSFT addresses mode collapse in SFT. An interesting parallel occurs in the reinforcement learning (RL) domain, where "entropy collapse" is also a core challenge—the model overfits to a few high-reward trajectories, thereby losing its exploratory capacity. I would like to ask the authors if they believe SSFT's core mechanism (i.e., using forking tokens and set matching to "protect" diversity) could offer new insights for solving entropy collapse in RL. Specifically, how might the ideas from SSFT be combined with or complement existing RL techniques aimed at maintaining policy entropy, such as entropy regularization?
- I've noticed that the experiments in this paper are primarily focused on mathematical reasoning datasets. A notable characteristic of these datasets is that a single problem often has multiple, distinct, and structurally varied correct solution paths. SSFT has achieved great results by using N explicit forking tokens to capture this diversity. However, my question is whether this advantage can naturally generalize to other, non-mathematical domains. In many general-purpose scenarios (such as open-ended question answering, creative writing, or code generation), "diversity" may manifest in a more implicit and subtle manner, rather than as the several clearly distinct "solutions" seen in math problems. In these contexts, is forcing a match between N discrete forking tokens and this more continuous or ambiguous diversity still the optimal strategy? Could the performance of SSFT in these general-purpose domains be limited by this "explicit forking" design?

---

> ### Author Response · Authors · 2025-11-25
> **Author Response #1 to Reviewer MjCW**
>
> We are truly grateful for your encouraging recognition of the novelty and elegance of the SSFT framework, as well as your appreciation of the rigor of our evaluations and presentation. We also greatly appreciate your thought-provoking ideas you shared.  New experimental results are summarized in the **``Summary of Concerns and New Experimental Results’’ section of the general response** and are also presented in a self-contained form in the **appendix of the latest revision.** We thank you for your thoughtful feedback and have done our best to address the identified weaknesses (**W#**) and questions (**Q#**).
>
> ## **W1: Implementation complexity. SSFT’s training process poses a barrier for larger models or datasets.**
> Although computing all $MN$ matching costs at full generation length can be expensive, we clarify in **Section 2.2**. that using a smaller matching length **L** can reduce such complexity.  Our ablation in **Figure 7 (Section 4)** shows that the $L=1000$ setting used throughout our experiments yields a significant performance gain while adding only $\frac{6.5h−6.3h}{6.3h}=3.2$\% training time in our setup. When additional compute time permits, L can be increased further and stopped once performance plateaus.
>
> ## **W2: Data Dependency. This distillation setting requires access to high-quality, diverse sets of reasoning paths.**
> In Section 4, “Robustness to Fine-Tuning Data Quality” (**Table 3**), and in the added “Robustness to Fine-tuning with code traces” (**Table 5**), we show that SSFT still improves over the baselines using only R1 traces. Since our method does not require explicit “diversity labels” and maximally prevents mode collapse from non-identical traces, we still observe empirical gains when all traces come from the same teacher model.  Additionally, we previously showed the final accuracy of R1 traces for s1 questions is around 62% in **Table 9 (Appendix 11.1)**. So SSFT can still show advantage over SFT even when the traces are not exceptionally high-quality or diverse.  However, we acknowledge that much larger-scale studies are needed to determine whether SSFT always outperforms SFT across all data quality and diversity settings.
>
> ## **W3: Heuristic for single-path inference.**
> Besides including more clarifications for our heuristic in **Appendix A.1 (Background)**, we apply a very small amount of RL to the distribution of global forking tokens. In **Sections 2.3 and 2.4**, we explain this Global Forking Policy Optimization (GFPO) and the new Pass@1 inference based on the distributions of forking tokens. We do not propose a new RL algorithm; rather, global forking tokens make additional RL extremely simple for learning a distribution that eliminates the need for a Pass@1 heuristic.
>
> Results for Pass@1 without the heuristic are denoted by “-GFPO” in **Table 1 (Section 3)** and **Table 4 (Appendix A.1)**. We also applied RL to the other baselines to ensure the comparisons remain fair throughout. The Cons@6 and Cons@32 results for SSFT-32B-GFPO are reported in **Table 1 (Section 3)**, showing that this additional RL step does not reduce performance under parallel test-time compute; in fact, it improves Cons@6 on AIME25 to 78.48%.

---

> ### Author Response · Authors · 2025-11-25
> **Author Response #2 to Reviewer MjCW**
>
> ## **Q1: Expected performance on much smaller (<7B) and larger models (>70B).**
> We added new experiments with Qwen3-4B-Base in **Table 6 (Appendix A.3)**. Given that the SSFT–SFT gap widens with model size (from 4B to 7B to 32B), we conjecture even larger gains at >70B.  We show Pass@1 improvements below, comparing SFT-mixed baselines to SSFT.
>
> | Pass@1 Acc | Improvement from SFT to SSFT using Qwen3-4B-Base | Improvement from SFT to SSFT using Qwen2.5-Math-7B-Instruct | Improvement from SFT to SSFT using Qwen2.5-32B-Instruct |
> |-----------|---------------------------------------------------|------------------------------------------------------------|---------------------------------------------------------|
> | AIME24    | 23.44 - 21.46 = 1.98                             | 51.25 - 46.15 = 5.1                                       | 64.06 - 58.23 = **5.83**                                    |
> | AIME25    | 23.13 - 21.98 = 1.15                             | 35.52 - 34.17 = 1.35                                      | 58.13 - 51.96 = **6.17**                                    |
> | MATH500   | 81.85 - 78.76 = 3.09                             | 89.74 - 86.62 = **3.12**                                      | 90.02 - 88.49 = 1.53                                    |
>
>
> ## **Q2: Could performance be improved by a mechanism that assigns an optimal N tailored to each specific question?**
> One potential idea is that we perform something like *top-*$p$ *sampling* to adjust $N$ at test time using the global forking distribution learned by GFPO. We can then adjust the number of votes, since Cons@$k$ may degrade beyond a certain $k$. Additional generations may not add diversity and can instead repeat the same mistakes.   We think your dynamic $N$ idea is very promising and could further improve the use of global forking tokens.
>
>
> ## **Q3: How might the ideas from SSFT be combined with existing RL techniques?**
> We think the new GFPO experiments added for **W3** provide a useful starting point for investigating such synergy. Your question on synergy with RL also inspired us to include the GFPO experiments. Two high-level ideas that come to mind are:
>
> **(a)** The diversity protected by SSFT can serve as a strong prior for exploration. During full RL, we can apply entropy regularization selectively.  e.g., to a subset of tokens such as global forking tokens, the forking tokens in [1], or low-entropy majority tokens, to prevent mode collapse while learning beyond the SSFT prior.
>
> **(b)** If diverse traces are unavailable for SSFT, an end-to-end RL alternative inspired by SSFT is to modify *Pass*@*k*-style training [2] so that, for the Pass@k reward, only one generation per global forking token is counted. This encourages diversity across forking paths rather than giving more trials for repeated samples from the same $g^{(i)}$.  Hopefully, diversity among randomly initialized global forking tokens can emerge from a ``set reward’’ design.
>
> [1] Wang, Shenzhi, et al. "Beyond the 80/20 rule: High-entropy minority tokens drive effective reinforcement learning for llm reasoning." arXiv preprint arXiv:2506.01939 (2025).
>
> [2] Chen, Zhipeng, et al. "Pass@ k training for adaptively balancing exploration and exploitation of large reasoning models." arXiv preprint arXiv:2508.10751 (2025).
>
> ## **Q4: Whether SSFT advantage generalizes to open-ended domains where “diversity” manifests implicitly.**
> We fine-tuned on only 1K code-generation questions (each with four traces) for the SSFT stage and 1.28K questions for the GFPO stage, and evaluated Pass@1 on LiveCodeBench (LCB). Full results, along with baselines, are added to **Table 2 (Section 4) and Table 5 (Appendix A.2)**. SSFT instills diverse, specialized reasoning modes **without collapse**, and RL can quickly leverage them to identify an optimal forking path and reasoning mode for each coding task, thereby significantly improving Pass@1. SSFT achieves 52.07% on LCBv5, which is 4.94% higher than the strongest baseline and 28.72% higher than the base model. SSFT coding model also shows better OOD generalization to math reasoning tasks.

---

### Author Response · Authors · 2025-11-25
**General Response by Authors (Part 3 of 3)**

### **3. Limited evaluation across model families and sizes. (reviewer `6BW4, Ut4D`)**
We added the same experiments using Qwen3-4B-Base, from the new Qwen3 family at the 4B scale; results are in **Table 6 (Appendix A.3)**. Our original manuscript includes Qwen2.5-32B-Instruct and Qwen2.5-Math-7B-Instruct. We plan to add Llama3.1-8B-Instruct in the next revision.  **Table 6** shows that SSFT remains robust on a smaller 4B model from the newer Qwen3 family.  However, we do observe a smaller gap than with the larger 32B and 7B models in **Tables 1 and 3**.

**Table 6 (Appendix A.3)**
|                              | AIME24 | AIME25 | MATH-500 | LCB(v5)* |
|------------------------------|-------:|-------:|---------:|--------:|
| **-------- Pass@1 with $g^{(i)}$ chosen by edge heuristic --------** |        |        |          |         |
| Qwen3-4B-Base                |   9.79 |   6.56 |   66.34  |   7.19  |
| SFT-mixed-distill-Qwen3-4B-tags | 21.46 | 21.98 |   78.76  |  **15.57**  |
| SSFT-Qwen3-4B (random σ)     |  23.33 | 21.87  |   79.63  |  12.57  |
| SSFT-Qwen3-4B                |  **23.44** | **23.13**  |  **81.85**  |  14.37  |
| **-------- Cons@6 --------** |        |        |          |         |
| Qwen3-4B-Base                |  13.03 | 10.91  |   78.63  |         |
| SFT-mixed-distill-Qwen3-4B-tags | 30.61 | 28.18 |   89.67  |         |
| SSFT-Qwen3-4B (random σ)     |  30.91 | **28.79**  |   89.73  |         |
| SSFT-Qwen3-4B                |  **32.42** | **28.79**  |   **91.20**  |         |
| **-------- Cons@32 --------**|        |        |          |         |
| Qwen3-4B-Base                |  20.00 | 16.67  |   83.00  |         |
| SFT-mixed-distill-Qwen3-4B-tags | 36.67 | 40.00 |   **91.80**  |         |
| SSFT-Qwen3-4B (random σ)     |  **43.33** | 36.67  |   91.60  |         |
| SSFT-Qwen3-4B                |  40.00 | **43.33**  |   **91.80**  |         |


### **4. Time complexity of SSFT (reviewer `MjCW`), why did you choose L=1000? (reviewer `Ut4D`, `8WvB`)**
Although computing all $MN$ matching costs at full generation length can be expensive, we clarify in **Section 2.2**. that using a smaller matching length $L$ can reduce such complexity.  Our ablation in **Figure 7 (Section 4)** shows that the $L=1000$ setting used throughout our experiments yields a significant performance gain while adding only $\frac{6.5h−6.3h}{6.3h}=3.2$\% training time in our setup. When additional compute time permits, L can be increased further and stopped once performance plateaus.

**Figure 7 (Section 4)**
|      Matching Length $L$                |   0   |  100  | 1000  | 2500  | 3500  |
|----------------------|------:|------:|------:|------:|------:|
| Training time (hours)|  6.30 |  6.32 |  6.50 |  7.00 |  7.40 |
| AIME25 Pass@1        | 55.10 | 55.42 | 58.40 | 59.02 | 58.95 |
| AIME25 Cons@32       | 80.00 | 80.00 | 86.66 | 83.33 | 86.66 |
| AIME24 Pass@1        | 61.77 | 62.19 | 64.06 | 63.87 | 63.95 |
| AIME24 Cons@32       | 80.00 | 80.00 | 83.33 | 86.66 | 83.33 |


We sincerely thank the reviewers for their careful evaluations and constructive feedback, which have substantially improved the rigor of this work.

---

### Author Response · Authors · 2025-11-25
**General Response by Authors (Part 2 of 3)**

## Summary of Concerns and New Experimental Results
We added **four new experimental results** to address the concerns and questions below. For ease of navigation, we present the results here using the **same section, table, and figure labels as in the paper**.  We also added **LiveCodeBench(v5)**, **LCB(v5)**, as part of evaluation of the model for out-of-distribution (OOD) generalization relative to the fine-tuning dataset, in response to the thoughtful suggestion from **reviewer `6BW4`**. These **OOD tasks** are marked with **an asterisk (*)** both in our main Table 1 and in the new results below.

### **1. Heuristic-based Selection of Global Forking Tokens for Single-Path Generation (reviewer `MjCW`, `8WvB`)**
Besides including more clarifications for our heuristic in **Appendix A.1 (Background)**, we apply a very small amount of RL to the distribution of global forking tokens. In **Sections 2.3 and 2.4**, we explain this Global Forking Policy Optimization (GFPO) and the new Pass@1 inference based on the distributions of forking tokens. We do not propose a new RL algorithm; rather, global forking tokens make additional RL extremely simple for learning a distribution that eliminates the need for a Pass@1 heuristic.

Results for Pass@1 without the heuristic are denoted by “-GFPO” in **Table 1 (Section 3) and Table 4 (Appendix A.1)**. We also applied RL to the other baselines to ensure the comparisons remain fair throughout. The Cons@6 and Cons@32 results for SSFT-32B-GFPO are reported in Table 1 (Section 3), showing that this additional RL step does not reduce performance under parallel test-time compute; in fact, it improves Cons@6 on AIME25 to 78.48%.

**Table 4 (Appendix A.1)**
| (Sampling $g^{(i)}$ for pass@1)                    | AIME24 | AIME25 | MATH-500 | LCB(v5)* |
|---------------------------------|-------:|-------:|---------:|--------:|
| SFT-mixed-distill-32B-tags-GRPO |  58.85 |  52.40 |    88.85 |   37.13 |
| SFT-mixed-distill-32B-tags-GFPO |  59.80 |  54.06 |    88.98 |   37.22 |
| SSFT-32B (random)-GFPO          |  59.58 |  53.96 |    89.87 |   36.53 |
| SSFT-32B-GFPO                   |  **64.22** |  **58.80** |    **89.90** |   **42.10** |


### **2. Does SSFT Provide Advantages in More Open-Ended Domains Such as Code Generation? (reviewer `MjCW`, `8WvB`)**
We fine-tuned on only 1K code-generation questions (each with four traces) for the SSFT stage and 1.28K questions for the GFPO stage, and evaluated Pass@1 on LiveCodeBench (LCB). Full results, along with baselines, are added to **Table 2 (Section 4)** and **Table 5 (Appendix A.2)**. SSFT instills diverse, specialized reasoning modes without collapse, and RL can quickly leverage them to identify an optimal forking path and reasoning mode for each coding task, thereby significantly improving Pass@1. SSFT achieves 52.07% on LCBv5, which is 4.94% higher than the strongest baseline and 28.72% higher than the base model. SSFT coding model also shows better OOD generalization to math reasoning tasks.

**Table 2 (Section 4),  Note: Cons@k for math tasks are in Table 5 (Appendix A.2)**
|  (Sampling $g^{(i)}$ for pass@1)  | LCB(v5) | AIME24* | AIME25* | MATH-500* |
|-----------------------------|--------:|--------:|--------:|---------:|
| Qwen2.5-32B-Instruct        |  23.35  |  15.80  |  10.40  |   80.40  |
| SFT-mixed-distill-32B-code  |  47.13  |  34.69  |  24.17  |   89.39  |
| SSFT-32B-code (random σ)    |  45.36  |  39.06  |  **32.92**  |   89.46  |
| SSFT-32B-code               |  **52.07**  |  **43.23**  |  32.82  |   **89.96**  |

---

### Author Response · Authors · 2025-11-25
**General Response by Authors (Part 1 of 3)**

Dear Reviewers,

We sincerely thank you for your interest in our method and for your thoughtful review of the paper. We appreciate your unanimous agreement that our SSFT approach, which frames parallel reasoning as set prediction, provides a novel framework for preventing reasoning mode collapse.  We highlighted our changes in blue in the latest revision and made the appendix as self-contained as possible to make navigation easier during this rebuttal.

In the sections below, we summarize the strengths highlighted by each reviewer, the main concerns they raised, and the additional experiments we conducted to address those concerns.

## Strengths Recognized by Reviewers

### **Novelty of our formulation and method**
- **Reviewer `MjCW`** -- appreciated the high innovation of our SSFT framework and global forking tokens for effectively mitigating mode collapse, a significant departure from existing fine-tuning paradigms; also praised the end-to-end self-supervised design as both elegant and highly effective.
- **Reviewer `6BW4`** -- noted a clear motivation from existing limitations of parallel scaling, and found the SSFT framework and matching design are technically sound.
- **Reviewer `Ut4D`** -- appreciated the usefulness and novelty of the set-of-next-token-prediction formulation, and the focus on an important question.
- **Reviewer `8WvB`** -- emphasized the novelty of the problem formulation and that we provide the first effective method for multi-teacher distillation, introducing global forking tokens as a powerful new form of steerable generation.

### **Rigorous execution and strong empirical results**
- **Reviewer `MjCW`** -- praised the thorough ablations on optimal matching components and SSFT’s superior performance on difficult reasoning benchmarks.
- **Reviewer `6BW4`** -- pointed out the improvements on both math and STEM reasoning tasks.
- **Reviewer `8WvB`** -- emphasized the rigor of our evaluation, noting that we compare against several baselines trained on the exact same data, and highlighted the significant gains in sample efficiency for parallel decoding.

### **Clear presentation and visualizations**
- **Reviewer `MjCW`** -- noted the logical, high-quality presentation, supported by intuitive and interpretable visualizations.
- **Reviewer `8WvB`** -- appreciated the visualization comparing thinking-token distributions, which powerfully confirms that SSFT prevents collapse; also emphasized that SSFT’s complex mechanism is made simple, intuitive, and self-contained in a single figure.

---

### Author Response · Authors · 2025-12-03
**General Response by Authors (Part 4 of 4)**

### **3 (Extra). Limited evaluation across model families and sizes. (reviewer `6BW4, Ut4D`)**

In our final revision, we additionally include Llama3.1-8B-Instruct as a base model for SSFT and all baselines, on top of the previously added Qwen3-4B-Base in "General Response by Authors (Part 3 of 3)" on Nov 25 and the Qwen2.5-32B-Instruct and Qwen2.5-Math-7B-Instruct models from the original submission.  The results are reported in **Table 7 (Appendix A.3)**.

**Table 7 (Appendix A.3)**
| Model                               | AIME 2024 Pass@1 | AIME 2024 Cons@32 | AIME 2025 Pass@1 | AIME 2025 Cons@32 | MATH-500 Pass@1 | MATH-500 Cons@32 |
|-------------------------------------|:----------------:|:-----------------:|:----------------:|:-----------------:|:---------------:|:----------------:|
| Llama3.1-8B-Instruct                | 3.65             | 13.33             | 0.94             | 6.67              | 47.53           | 65.40            |
| SFT-mixed-distill-Llama3.1-8B       | 4.90             | 13.33             | 4.90             | 10.00             | 61.95           | **80.00**        |
| SSFT-Llama3.1-8B (random σ)         | 6.77             | 13.33             | 4.79             | 10.00             | **62.22**       | 79.60            |
| SSFT-Llama3.1-8B                    | **6.98**         | **16.66**         | **5.79**         | **20.00**         | 62.04           | **80.00**        |

Once again, we observe that SSFT yields consistent improvements across all three math reasoning tasks for the Llama-3.1 family. Although the performance after fine-tuning Llama3.1-8B-Instruct as the base model is noticeably lower than with the Qwen models, this pattern is consistent with recent work. The overall low scores therefore reflect the weaker pretrained base model rather than limitations of the post-training algorithm or implementation. Compared to recent post-training papers, our results are comparable to, and in some cases, such as on the harder AIME25 benchmark, surpass RL-based methods that require substantially more compute.

(pass@1 accuracy)
- [3] reports 1%  on DAPO-Math-17K dataset during RL fine-tuning with Llama3.1-8B-Instruct model.
- HICRA [4] applies RL to Llama3.1-8B-Instruct on the DAPO-Math-17K DeepScaleR dataset, which requires much more compute, and achieves 8.3% on AIME24, 0.8% on AIME25, and 54.8% on MATH-500.
- Pass@K RL[5] fine-tunes Llama3.1-8B-Instruct and achieves 8.7% on AIME24 and 0.9% on AIME25.
- S1.1 [1] and Multiverse [2] only experiment with Qwen2.5-32B-Instruct as the base model.

[1] Muennighoff, Niklas, et al. "s1: Simple test-time scaling." Proceedings of the 2025 Conference on Empirical Methods in Natural Language Processing. 2025.

[2] Yang, Xinyu, et al. "Multiverse: Your Language Models Secretly Decide How to Parallelize and Merge Generation." arXiv preprint arXiv:2506.09991 (2025).

[3] Wang, Shenzhi, et al. "Beyond the 80/20 rule: High-entropy minority tokens drive effective reinforcement learning for llm reasoning." arXiv preprint arXiv:2506.01939 (2025).

[4] Wang, Haozhe, et al. "Emergent hierarchical reasoning in llms through reinforcement learning." arXiv preprint arXiv:2509.03646 (2025).

[5] Chen, Zhipeng, et al. "Pass@ k training for adaptively balancing exploration and exploitation of large reasoning models." arXiv preprint arXiv:2508.10751 (2025).

---

### Meta-Review · Area_Chair_exXC · 2026-01-08

**Summary:**

This paper proposes Set Supervised Fine-Tuning (SSFT), a novel framework that formulates parallel reasoning as a set prediction problem using global forking tokens and bipartite matching to prevent reasoning mode collapse. Reviewers broadly agreed on the novelty, technical soundness, and clarity of the approach, as well as the strong empirical gains on challenging math and reasoning benchmarks. Initial concerns centered on (i) limited evaluation across model families and scales, (ii) computational complexity and scalability of the matching procedure, (iii) reliance on heuristics for single-path (Pass@1) inference, and (iv) generalization beyond math reasoning to more open-ended domains. The authors provided an extensive rebuttal with substantial new experiments, ablations, and clarifications. These additions significantly strengthened the empirical foundation of the work and addressed most of the reviewers’ concerns.

The paper presents a clearly novel and well-motivated contribution with strong empirical validation. The authors’ rebuttal substantially improves the work by addressing nearly all major reviewer concerns through additional experiments, ablations, and methodological refinements. While some open questions remain regarding extreme-scale models and highly open-ended tasks, these are reasonable limitations for the current scope and do not detract from the core contribution. The paper should be of broad interest to the ICLR community working on reasoning, distillation, and fine-tuning paradigms.

**Reviewer Concerns:**

**Concerns largely addressed by the rebuttal:**

Limited evaluation across model families and sizes:
The authors added results on Qwen3-4B-Base and Llama3.1-8B-Instruct, in addition to the original Qwen2.5-7B and 32B models. These experiments demonstrate that SSFT consistently outperforms SFT-style baselines across families and scales, with larger gains at higher model sizes.

Out-of-distribution and non-math generalization:
New experiments on LiveCodeBench (LCB v5), including both math→code and code→math transfer, show that SSFT improves OOD generalization and is effective beyond purely mathematical reasoning.

Heuristic for Pass@1 inference:
The introduction of Global Forking Policy Optimization (GFPO) replaces the heuristic with a lightweight RL-based approach, improving Pass@1 while preserving Cons@k performance.

Matching length and computational complexity:
Ablations on the matching length parameter (L) demonstrate that the chosen setting achieves a strong trade-off between efficiency and performance, alleviating concerns about scalability.

Fairness of baselines:
The authors incorporated improved SFT-mixed baselines with distinct tags, addressing concerns about unfair comparisons.

**Concerns partially or still outstanding:**

Scalability to very large models:
While the authors provide a reasonable conjecture based on scaling trends, empirical validation at larger scales remains future work.

Data dependency and availability of diverse traces:
The rebuttal shows robustness to lower-quality and less diverse traces, but large-scale evidence across varied data regimes is still limited.

Applicability to highly open-ended or creative tasks:
Code generation results are promising, but the effectiveness of explicit forking tokens in domains with less well-defined correctness remains an open question.

**Reviewer Scores:**

Reviewer MjCW:
Likely to keep a positive score 6, given that the main weaknesses (Pass@1 heuristic, scalability concerns, and domain generalization) were directly addressed through GFPO, extensive ablations, and new code-generation experiments. The reviewer already expressed strong enthusiasm for the core idea, and the rebuttal substantially strengthens confidence in the method.

Reviewer 6BW4:
Likely to keep a positive score 6, as requests for broader evaluation across model families and sizes, as well as OOD generalization analysis, were fully addressed with Qwen3-4B and Llama3.1-8B results and LiveCodeBench experiments.

Reviewer Ut4D:
Likely to remain at 6, as concerns about robustness, matching behavior, and comparisons to RL-based methods were addressed but may not fully resolve all skepticism.

Reviewer 8WvB:
Already increased soundness (3 → 4) during discussion; overall score would likely remain at 6, but with stronger confidence leaning toward acceptance after the added ablations (matching length), GFPO, and expanded task scope.

---

### Decision · Program_Chairs · 2026-01-26

Accept (Poster)